

# Potential of polarization/Raman lidar to separate fine dust, coarse dust, maritime, and anthropogenic aerosol profiles

Rodanthi-Elisavet Mamouri[1,2] and Albert Ansmann[3]

[1]Cyprus University of Technology, Dep. of Civil Engineering and Geomatics, Limassol, Cyprus
[2]The Cyprus Institute, Energy, Environment, and Water Research Center, Nicosia, Cyprus
[3]Leibniz Institute for Tropospheric Research, Leipzig, Germany

*Correspondence to:* R. E. Mamouri (rodanthi.mamouri@cut.ac.cy)

**Abstract.** We applied the recently introduced Polarization Lidar Photometer Networking (POLIPHON) technique for the first time to triple-wavelength polarization lidar measurements at 355, 532, and 1064 nm. The lidar observations were performed at Barbados during the Saharan Aerosol Long-Range Transport and Aerosol-Cloud-Interaction Experiment (SALTRACE) in the summer of 2014. POLIPHON comprises the traditional lidar technique to separate mineral dust and non-dust backscatter con-

tributions and the new extended approach to separate even the dust backscatter component into fine and coarse dust fractions. We show that the traditional and the extended methods are compatible and lead to the same set of dust and non-dust profiles at simplified, less complex aerosol layering and mixing conditions as is the case over the remote tropical Atlantic. To derive dust mass concentration profiles from the lidar observations, trustworthy extinction-to-volume conversion factors are needed and obtained from an updated, extended AERONET sun photometer data analysis of the correlation of fine, coarse and total

dust volume concentration with dust extinction coefficients for all three laser wavelengths. Conversion factors for pure marine aerosol conditions and continental anthropogenic aerosol situations are presented in addition. As an additional new feature of POLIPHON, the Raman-lidar method for particle extinction profiling is used to identify the aerosol type (marine or anthropogenic) of the non-dust aerosol fraction. The full POLIPHON methodology was successfully applied to a SALTRACE case and the results are discussed. We conclude that the 532 nm polarization lidar technique has many advantages in comparison to

respective 355 and 1064 nm polarization lidar approaches and leads to most robust and accurate POLIPHON products.

## 1 Introduction

Polarization lidar is a very powerful remote sensing tool for aerosol and cloud research. The technique has been used since a long time to monitor and investigate cirrus cloud systems (e.g., Sassen, 1991, 2005; Reichardt et al., 2002, 2008) and polar stratospheric cloud evolution (see, e.g., Browell et al., 1990; Achtert and Tesche, 2014). The technique is well suited to study

heterogeneous ice formation in mixed-phased clouds (e.g., Sassen et al., 2003; Ansmann et al., 2005, 2008; Ansmann et al., 2009a; Seifert et al., 2010, 2011) and liquid-water cloud developments (e.g., Bissonnette, 2005; Donovan et al., 2015). The technique is meanwhile intensively used to explore aerosol mixtures, and here especially for soil, desert, and volcanic dust identification (e.g., McNeil and Carswell, 1975; Iwaska and Hayashida, 1981; Winker and Osborn, 1992; Gobbi, 1998; Murayama et al., 1999; Murayama et al., 2004; Cairo et al., 1999; Gobbi et al., 2000; Sakai et al., 2003; Sassen et al., 2007; Freudenthaler et al.,





2009; Ansmann et al., 2010; Ansmann et al., 2011a; Groß et al., 2012; Miffre et al., 2012; Amiridis et al., 2013), and allows us to unambiguously discriminate desert dust or volcanic dust from other aerosols such as biomass-burning smoke, maritime particle, or urban haze (e.g., Sugimoto et al., 2003; Shimizu et al., 2004; Nishizawa et al., 2007; Tesche et al., 2009a; Ansmann et al., 2012). Another fruitful field is related to aerosol typing which is based on combined data sets of parti-

cle extinction-to-backscatter (lidar ratio) obtained with Raman lidar or High Spectral Resolution Lidar (HSRL) and particle linear depolarization ratio obtained with the polarization lidar (e.g., Burton et al., 2012, 2013; Groß et al., 2013, 2015; Illingworth et al., 2015). Recently, we broadened the application spectrum of polarization lidar technique by introducing the POLIPHON (Polarization Lidar Photometer Networking) method for estimating fine dust (particles with radius <500 nm) and coarse dust mass concentration profiles from single-wavelength polarization lidar measurements (Mamouri and Ansmann,

2014). Furthermore, we demonstrated that polarization lidar has the potential to contribute to vertical profiling of cloud condensation nucleus (CCN) and ice-nucleating particle (INP) concentrations (Mamouri and Ansmann, 2015, 2016).

The retrieval of these higher level lidar products, that are useful for environmental/health and climate impact studies, is driven by the fact that state-of-the-art global atmospheric models meanwhile include aerosol schemes which simulate microphysical aerosol processes and life cycles of a variety of aerosol components and can be used to predict dust events and enable even

the simulation of CCN and INP profiles (Huneeus et al., 2011; Koffi et al., 2012, 2016; Mann et al., 2014; Kim et. al., 2014; Nickovic et al., 2016; Hande et al., 2016). These aerosol modeling and forecast activities need a strong accompanying observational (profiling) component to guarantee high quality of the simulation products. In this respect, lidar plays an important role as the optimum aerosol profiling technique. The increasing potential of present and future atmospheric modeling makes it necessary to explore to what extend lidar can support numerical modeling by providing observations of the most relevant

aerosol parameters.

The attempt to separate even fine dust and coarse dust contributions to the total dust load is motivated by the fact that air-quality management needs quantitative information on the contribution of natural fine dust particles to the measured fine-particle concentration ($PM_1$, particles with diameters <1 $\mu$m), especially in areas close to deserts, to get a better insight into the natural impact on recorded high and unhealthy aerosol pollution levels in terms of $PM_1$, $PM_{2.5}$ (particle diameter <2.5 $\mu$m),

and $PM_{10}$ (particle diameter <10 $\mu$m). Lidar-based fine and coarse dust separation is also of interest to evaluate basic dust emission parameterizations in dust transport models (Ansmann et al., 2017). Removal of dust by gravitational settling depends on the simulated ratio of fine-to-coarse dust particle number concentrations. Fine and coarse dust fractions also show different radiative influences (Nabat et al., 2012; Ridley et al., 2016), and modeled CCN and INP concentration may be rather erroneous when the ratio of the fine-to-coarse dust particle number concentration and the underlying simulated size distribution deviate

significantly from the lidar measurements.

The advantage of the POLIPHON method is that this technique does not need a critical dust particle shape model in the data analysis of observed dust episodes, as required in alternative lidar-photometer retrieval schemes (Chaikovsky et al., 2012; Lopatin et al., 2013; Torres et al., 2016). Another advantage is that the POLIPHON is based on single-wavelength lidar observations, thus the method is likewise simple and therefore robust and can be easily automatized and applied to continuously

measured large data sets. An important advantage is that the method can be used even at cloudy conditions. The alternative





lidar-photometer approaches need clear skies and even absolutely cirrus-free conditions which are often not given during dust outbreak events. The robust POLIPHON method is thus well suited for environmental studies and aerosol-cloud interaction research. Since POLIPHON can provide results at any atmospheric conditions (accept during rain periods), unbiased statistics on aerosol mixture conditions can be gained from long-term observations as well.

In this article, we extend the methodology for fine and coarse dust separation developed for 532 nm towards 355 and 1064 nm lidar wavelengths. A dense set of high-quality observational data obtained with a triple-wavelength polarization lidar collected in the framework of Saharan Aerosol Long-Range Transport and Aerosol-Cloud-Interaction Experiment (SALTRACE) (Haarig et al., 2016a, 2017a) is available for this study. We are able to test our POLIPHON method for the first time simultaneously for the three basic aerosol lidar wavelengths of 355, 532, and 1064 nm. The importance of such a multi-wavelength

effort is corroborated by the fact that NASA's spaceborne polarization lidar CALIOP (Cloud Aerosol Lidar with Orthogonal Polarization) (Winker et al., 2009) is operated at 532 nm, whereas ESA's future spaceborne aerosol lidar ATLID (Atmospheric Lidar) will be operated at 355 nm (Illingworth et al., 2015), and to homogenize these different polarization lidar data sets a good knowledge about the compatibility of 355 and 532 nm retrieval schemes is required. In this article, we will discuss which of the available three wavelengths is the optimum one and produces the most robust and accurate retrieval data sets when using

POLIPHON.

SALTRACE provided excellent conditions to test and check different POLIPHON retrieval concepts. Over Barbados (during the summer seasons), we observed well-defined almost pure dust conditions in the lofted dust layers between 1.5 and 5.5 km height, and also well-defined marine and dust particle mixing states below the Saharan air layer (SAL). At these conditions the consistency of results obtained by applying different approaches (by the so called one-step and two-step methods, explained in

Sect. 4 ) could be tested in detail.

Another favorable aspect to further explore the applicability of the POLIPHON method is that recently the results of a comprehensive laboratory study on particle linear depolarization ratios for a variety of soil and desert dust types were published (Järvinen et al., 2016). The most interesting point is that all measurements were performed as a function of dust particle size. In this way, for the first, very detailed information on fine dust depolarization ratios became available and can be used in the

POLIPHON methodology which was based before on a few laboratory measurements of fine and coarse-dust-dominated linear depolarization ratios (Sakai et al., 2010) and simulation studies by Gasteiger et al. (2011). Recent new field observations in advected thick dust plumes over Senegal in western Africa (Veselovskii et al., 2016) and Tajikistan in central Asia (Hofer et al., 2017) corroborate the laboratory findings. In the field of modeling, more studies became available (Kemppinen et al., 2015a, b) presenting results in terms of dust lidar ratio and depolarization ratio for irregularly shaped particles as a function of size.

We will review the compiled information on recent dust depolarization model, laboratory, and field studies in Sect. 2.

The paper contains six sections. In Sect. 3, the SALTRACE field campaign, the lidar observations, and the Aerosol Robotic Network (AERONET) data sets used in this article are introduced and briefly described. The theoretical background of the POLIPHON method is outlined in Sect. 4. Particle extinction-to-volume conversion factors for the used laser wavelengths and basic aerosol types (marine, continental pollution, desert dust) are required to retrieve mass concentration profiles. These

factors are quantified on the basis of multiyear and short-term field campaign AERONET observations. The main results are



presented in Sect. 5. An extended case study of a POLIPHON data analysis is then given for a SALTRACE measurement in the summer of 2014 (Sect. 6). All available POLIPHON approaches (one-step and two-step POLIPHON and combination of both) are applied and compared, and the POLIPHON method is applied to all three laser wavelengths. Concluding remarks are given in Sect. 7.

## 2 Review of recent field, laboratory and modeling studies of dust-related particle linear depolarization ratio

Järvinen et al. (2016) measured the particle linear depolarization ratio in a large chamber filled with natural probes of dust from different deserts in Africa, Asia, and America. They were able to inject dust particles of a specific size class into the chamber and thus to perform observations as a function of the size parameter ($\pi D/\lambda$ with laser wavelength $\lambda$ and particle diameter $D$). The studies were performed at a wavelength of 488 nm, partly also at 552 nm, and consider dust particles up to size parameters of 20 (and thus up particle diameters of about 3 $\mu$m). From this work, we estimated the particle linear depolarization for fine dust for all three wavelengths. These values are given in Table 1. Fine dust causes particle depolarization ratios around 0.21±0.02 , 0.16±0.02, and 0.09±0.03 for the laser wavelengths of 355, 532, and 1064 nm, respectively. These values are in agreement with the laboratory findings of Sakai et al. (2010) who found dust particle depolarization ratios of $0.14-0.17\pm0.03$ for fine-dust-dominated particle ensembles at 532 nm. The fine dust depolarization ratio for 532 nm is also in good agreement with modeling studies (Gasteiger et al., 2011). However, conclusions on the depolarization ratio for atmospheric coarse dust size distributions cannot be drawn. Järvinen et al. (2016) stated that gravitational settling in the chamber had a strong impact on their measurement with large dust particles so that most reliable results are available for dust size parameters <10 only.

Table 1 also contains the coarse dust depolarization ratios we use in our POLIPHON method. The values are derived from both laboratory and field observations. The laboratory studies of Sakai et al. (2010) yielded particle linear depolarization ratios of 0.39 at 532 nm for coarse dust ensembles (Asian and Saharan dust probes). Burton et al. (2015) reported such high particle linear depolarization ratios around 0.4 for 532 and 1064 nm in dense dust plumes over the United States close to the ground and probably close to the dust sources with a very dominating coarse mode. Veselovskii et al. (2016) and Hofer et al. (2017) found maximum particle depolarization ratios around 0.35 at 532 nm, respectively, in dense western African and central Asian dust plumes. The accompanying AERONET sun photometer observations indicated an extinction-related fine-mode fraction (FMF) of 10–15% (at 500 nm) so that the coarse dust depolarization ratio is again close to 0.39-0.4 at such conditions when assuming a fine dust depolarization ratio of, e.g., 0.16. Here we have to assume that FMF is equal to the backscatter-related FMF, denoted as BFMF, which is the relevant parameter for lidar applications. BFMF=FMF suggests that the extinction-to-backscatter ratio (lidar ratio) for fine and coarse dust particles is equal. Models suggest that the lidar ratio for fine dust is almost a factor of two higher than the one for coarse dust (Gasteiger et al., 2011; Kemppinen et al., 2015a, b) which would lead to BFMF values a factor of 2 lower than FMF. Our combined polarization-lidar AERONET-photometer observations are not consistent with a difference of a factor of two between the fine and coarse dust lidar ratio. The same can be concluded from the study of Veselovskii et al. (2016). For simplicity, and as long as our knowledge is poor in this respect, we suggest that BFMF is approximately equal to FMF, and consequently, the fine dust and coarse dust lidar ratios are equal.



It should be emphasized that the combination of light depolarization and extinction observations provides unique and complementary information to separate fine and coarse dust fractions. In practice, we can use the AERONET FMF information to adjust the coarse dust depolarization ratio to obtain an overall consistent set of measured total and assumed fine dust and coarse dust depolarization ratios.

In contrast to the comparably high total dust linear depolarization ratio of around 0.3 at 532 nm (after dust long-range transport) the 1064 nm dust depolarization ratio was found to be in the range of 0.22–0.28 for long-range-transported dust (Freudenthaler et al., 2009; Burton et al., 2015; Haarig et al., 2017a). According to AERONET observations the FMF for the 1020 nm aerosol channel is typically in the range of 0.05–0.08 at dusty conditions. We can conclude that the 1064 nm dust depolarization ratio is almost completely controlled by coarse dust. The maximum values for the 1064 nm total dust depolar-

ization ratio measured over Barbados of 0.27 are almost equal to the coarse dust depolarization ratios of 0.28 (Haarig et al., 2017a), when assuming a BFMF of 0.06 at 1064 nm. Even over Morocco, maximum 1064 nm depolarization ratios were significantly below 0.3 with values of 0.28 (Freudenthaler et al., 2009). Only during events close to dust emissions zones when large to giant dust particles and even sand particle (having diameters >60 $\mu$m) occur, high particle linear depolarization ratios (up to 0.4) are obviously measurable at 1064 nm (Burton et al., 2015). This is corroborated by 710 nm depolarization ratio

studies during dust devil events in Morocco (Ansmann et al., 2009a).

We can only speculate about the reason for the significantly different coarse dust depolarization ratio at 1064 nm (<0.3) and 532 nm (0.35-0.39). Simulation studies of Kemppinen et al. (2015a) show a possible way of explanation. Their simulations are based on realistic dust particle shapes sampled during the Saharan Mineral Dust Experiment, SAMUM-1, in Morocco (Lindqvist et al., 2014). For the so-called dolomite shape type, the simulations yield particle linear depolarization ratios of 0.2–

0.25 and 0.35–0.4, respectively, at 532 and 1064 nm in the case of dust particles with diameters around 2 $\mu$m. It is worthwhile to mention in this context, that the AERONET photometer observations during dust outbreak over Barbados in 2013 and 2014 show that the coarse-mode effective diameter accumulates around 3$\pm$0.4$\mu$m which indicates that most coarse dust particles after long-range transport have diameters in the 1-3 $\mu$m size range. Veselovskii et al. (2016) retrieved overall (fine dust + coarse dust) effective diameters of 2–2.5 $\mu$m from the multiwavelength lidar measurements in the dust plumes over Senegal during

the SHADOW campaign in March and April 2015. Aircraft observations (in situ aerosol measurements) over Cabo Verde and Barbados indicate a pronounced dust size mode around 1-2 $\mu$m and a drop by an order of magnitude in number concentration from from diameters of 2 $\mu$m to 4-5 $\mu$m (Weinzierl et al., 2017).

The dust linear depolarization ratio at 355 nm seems to be always close to 0.25 (Groß et al., 2011, 2015; Burton et al., 2015; Haarig et al., 2017a), disregarding the distance from the dust source and strength of the outbreak, and occuring particle sizes.

Even during rather strong dust storms in central Asia, Hofer et al. (2017) found maximum 355 nm depolarization ratios of 0.29 only. AERONET sun photometer observations typically point to fine-mode fractions of 0.3-0.5 for the aerosol channel centered at 380 nm during dust events. Obviously, the 355 nm depolarization ratio is always strongly influences by submicrometer dust particles. If we now consider a fine dust linear depolarization ratio of 0.21 according to the laboratory study of Järvinen et al. (2016) and a fine-mode backscatter fraction of 0.5 we end up with a coarse dust depolarization ratio in the range of 0.27-0.3

which is in agreement with the observations of 0.29 of Hofer et al. (2017).



## 3 Experimental data

### 3.1 SALTRACE field campaign

The SALTRACE field studies were conducted at Barbados in June-July 2013 (SALTRACE-1), February-March 2014 (SALTRACE-2), and in June-July 2014 (SALTRACE-3). An overview of the SALTRACE field activities and goals can be found in Weinzierl et al.
(2017) and (Haarig et al., 2017a). As a unique contribution, triple-wavelength polarization lidar observations of lofted dust layers were performed at the Caribbean Institute of Meteorology and Hydrology (CIMH, 13.1°N, 59.6°W, 90 m above sea level) about 5000-8000 km downwind the main Saharan dust source regions. We selected a measurement conducted in June 2014 to demonstrate the potential of the extended POLIPHON method.

### 3.2 Triple-wavelength polarization lidar

The containerized multiwavelength polarization/Raman lidar BERTHA (Backscatter Extinction lidar-Ratio Temperature Humidity profiling Apparatus) (Althausen et al., 2000; Haarig et al., 2017a) was involved in 12 major aerosol-related field campaigns in Europe, Asia, and Africa during the last 20 yeaers. The advanced lidar was re-designed in 2012 to allow particle linear depolarization measurements at 355, 532, and 1064 nm, simultaneously. The fundamental, basic quantities measured with lidar and input in the POLIPHON retrieval are the particle linear depolarization ratio and the particle backscatter coeffi-
cient for a given wavelength. The determination of these basic aerosol parameters from polarization-sensitive BERTHA signals is described by Haarig et al. (2017a). Furthermore, the case study used here as the POLIPHON demonstration case in Sect. 6, is described in detail in terms of the aerosol optical properties by Haarig et al. (2017a), including a discussion of the uncertainties in the derived aerosol optical properties.

### 3.3 AERONET sun photometry

An important prerequisite of the POLIPHON methodology is the existence of a close relationship between the lidar-derived particle extinction coefficient and the particle volume concentration, which allows us to compute the desired mass concentration profiles. We need these extinction-to-volume conversion factors for the basic aerosol types, i.e., for marine, anthropogenic, and dust aerosols, and in the case of dust even separately for fine dust and coarse dust. The correlation between particle extinction coefficient and particle volume concentration is intensively studied in Sect. 5 based on long-term AERONET sun photometer
observations and measurements during dedicated field campaigns such as SAMUM-1 (Morocco, 2006), SAMUM-2 (Cabo Verde, 2008), and SALTRACE (Barbados, 2013-2014) (AERONET, 2016). The study makes use of the same AERONET data sets for desert, marine, and polluted continental sites as described already in detail by Mamouri and Ansmann (2016) in their study on CCN and INP profiling with polarization lidar.

Fourteen years of AERONET observations at Leipzig, Germany, performed by the Leibniz Institute for Tropospheric Re-
search (TROPOS) from 2001-2015 and four years of AERONET observations at Limassol, Cyprus, performed by the Cyprus University of Technology (CUT) from 2011 to 2015 (Nisantzi et al., 2014, 2015) are available for our studies. Aerosol mixtures




of anthropogenic haze, biomass burning smoke, soil and road dust, and marine particles, and strong dust outbreaks from Middle East deserts and the Sahara frequently occur over Cyprus (Nisantzi et al., 2015). Pure dust observations are available from the Saharan Mineral Dust Experiments SAMUM-1 (Ouarzazate, Morocco) (Toledano et al., 2009) and SAMUM-2 (Praia, Cabo Verde) (Toledano et al., 2011), and SALTRACE (Weinzierl et al., 2017). These dust AERONET data sets allow us to study

the extinction-to-volume correlation separately for fine, coarse, and total dust. Furthermore, we used 7.5 years of data from the AERONET station at Ragged Point, Barbados (Prospero and Mayol-Bracero, 2013) to study the correlation between the aerosol extinction coefficient and volume concentration for pure marine conditions. An overview of the observational periods and amount of available data for the analyzed different aerosol conditions with focus on the three defined aerosol types can be found in Table 1 in Mamouri and Ansmann (2016). More details of these AERONET stations are given on the AERONET

web page (http://aeronet.gsfc.nasa.gov). Dubovik et al. (2000) carried out a detailed analysis of uncertainties in the AERONET products. Errors in the basic measurements of the aerosol optical thickness (AOT) are low (of the order of 0.01) and about 10–20% for the retrieved particle volume concentrations.

## 4 POLIPHON method

The POLIPHON concept (Mamouri and Ansmann, 2014) was introduced to separate dust and non-dust aerosol components

and to estimate, in addition, the fine and coarse dust contributions to the overall backscatter and extinction coefficients and particle mass concentration. Of key importance is the avoidance of a particle shape model for the irregularly shaped mineral dust particles in the retrieval scheme. The method is solely based on the use of characteristic depolarization ratios for fine dust, coarse dust, and non-dust aerosol. In Sect. 2, field observations and available laboratory studies of these required desert dust depolarization ratios were reviewed. An overview of the data analysis scheme is given in Fig. 1. All retrieval steps are

explained in Sects. 4.1–4.5.

We start with a brief overview about the full methodology. The POLIPHON approach comprises both the traditional retrieval (here denoted as one-step method when using a polarization lidar for dust/non-dust separation) and the recently introduced so-called two-step POLIPHON method (Mamouri and Ansmann, 2014). As presented in Sect. 4.2, the one-step method permits the separation of the dust and non-dust components in one step, but no further separation into fine and coarse dust fractions.

This method is the classical, well-established way of dust separation by means of the measured particle linear depolarization ratio (e.g., Sugimoto et al., 2003; Shimizu et al., 2004; Tesche et al., 2009a). In contrast, two steps are required to separate non-dust, fine dust, and coarse dust components (Sect. 4.1). In the first step, the coarse dust fraction is isolated. In the second step, the remaining fine dust plus non-dust backscatter is analyzed and the fine dust and non-dust fractions are separated. The different retrieval schemes are illustrated in Fig. 1 of Mamouri and Ansmann (2014). The consistency between the one-step

and the two-step method is discussed in Sect. 4.6 and demonstrated in Sect. 6 (SALTRACE case study).

To convert the obtained aerosol-type-dependent backscatter coefficients into respective extinction coefficients, characteristic lidar ratios for fine dust, coarse dust, and the non-dust component are required (Sect. 4.3). These extinction-to-backscatter ratios are taken from the literature. Although simulations suggest different lidar ratios for fine and coarse dust (Gasteiger et al.,





2011; Kemppinen et al., 2015a, b), we assume just the same lidar ratio for fine, coarse, and total dust. This assumption is corroborated by the consistency between dust observations with polarization lidar (providing indirectly information on the fine-mode-to-coarse-mode backscatter ratio) and sun photometer (providing information on the fine-mode-to-coarse-mode extinction ratio), when assuming that the fine dust and coarse dust depolarization ratios are approximately equal.

As a new aspect of the extended POLIPHON method, we make use of the Raman-lidar solution for the overall particle extinction coefficient (see Haarig et al., 2017a, for more details of this retrieval) to identify the non-dust aerosol type (marine or continental anthropogenic or even a mixture of both, see Sect. 4.4, also indicated in Fig. 1). Finally, extinction-to-volume conversion factors obtained from an extended AERONET data analysis (presented in Sect. 5) are used together with assumed particle densities (for marine, anthropogenic, and dust particles) to obtain particle mass concentration profiles for fine dust

($PM_{1,d}$), coarse dust ($PM_{10,d} - PM_{1,d}$), total dust ($PM_{10,d}$), anthropogenic aerosol ($PM_{10,c}$, usually fine-mode particles only, $PM_{1,c}$), and marine aerosol ($PM_{10,m}$), according to the equations presented in Sect. 4.5.

## 4.1    One-step POLIPHON method

Because a detailed description of the basic POLIPHON data analysis procedure is given in Mamouri and Ansmann (2014), we only present the set of equations needed in the data analysis starting from the measured height profiles of the particle linear

depolarization ratio $\delta_p$ and particle backscatter coefficient $\beta_p$. In the one-step approach, the dust backscatter coefficient $\beta_d$ and the non-dust backscatter coefficient $\beta_{nd}$ are obtained with the following set of equations:

$$\beta_d = \beta_p \frac{(\delta_p - \delta_{nd})(1 + \delta_d)}{(\delta_d - \delta_{nd})(1 + \delta_p)} \text{ for } \delta_{nd} < \delta_p < \delta_d, \tag{1}$$

$$\beta_d = \beta_p \text{ for } \delta_p \geq \delta_d, \tag{2}$$

$$\beta_d = 0 \text{ for } \delta_p \leq \delta_{nd}, \tag{3}$$

$$\beta_{nd} = \beta_p - \beta_d. \tag{4}$$

The assumed quantities are the overall (fine + coarse) dust depolarization ratio $\delta_d$ and and the non-dust depolarization ratio $\delta_{nd}$. In Sect. 6, we use total dust depolarization ratios of $\delta_d$ of 0.25 (355 nm), 0.31 (532 nm), and 0.27 (1064 nm) (see Table 1). For $\delta_{nd}$ we assume 0.05 considering minor contributions to depolarization by dried marine particles (Haarig et al., 2017b) or by anthropogenic particles.

## 4.2    Two-step POLIPHON method

As mentioned above, two subsequent steps of computations are required to obtain height profiles of non-dust, fine dust, and coarse dust backscatter height profiles. In step 1, we separate the height profile of the coarse dust backscatter fraction from the remaining aerosol backscatter caused by non-dust and fine dust particles. We assume that coarse dust produces a linear depolarization ratio of, e.g., $\delta_{dc}$ of 0.39 at 532 nm, 0.27 at 355 nm, and 0.28 at 1064 nm (see Table 1, and Sect. 6).

As an important task in the first round, we have to estimate the depolarization ratio for the remaining aerosol (mixture of non-dust particles and fine dust). We may assume, e.g., that in the SAL over Barbados 33% of the backscatter of the remaining



aerosol is caused by marine particles, producing an assumed depolarization ratio of 0.05 at 532 nm, and that 67% of the backscatter of the remaining aerosol is caused by fine dust. At such conditions the depolarization ratio of the remaining aerosol is $\delta_{\mathrm{nd+df,e}}=0.12$ according Eq. (2) in Mamouri and Ansmann (2014) when using $\beta_{\mathrm{df}}$ and $\delta_{\mathrm{df}}$ instead of $\beta_{\mathrm{d}}$ and $\delta_{\mathrm{d}}$ in that equation. The index e indicates that this parameter is estimated in the first round. Note, that such an input parameter is not needed in

the traditional retrieval (one-step method) so that comparisons with the results of the one-step method (later on in Sect. 4.6) are used to evaluate how reliable this attempt to estimate the residual aerosol depolarization ratio is. The solutions profiles of the one-step and the two-step methods can even be used to estimate the optimum height profile for $\delta_{\mathrm{nd+df}}$ as will be shown in Sect. 6.

The coarse dust backscatter coefficient $\beta_{\mathrm{dc}}$ and the backscatter coefficient $\beta_{\mathrm{nd+df}}$ of the the remaining aerosol mixture are

now obtained by means of the estimated input value $\delta_{\mathrm{nd+df,e}}$ and the following set of equations (Mamouri and Ansmann, 2014):

$$\beta_{\mathrm{dc}} = \beta_{\mathrm{p}} \frac{(\delta_{\mathrm{p}} - \delta_{\mathrm{nd+df,e}})(1 + \delta_{\mathrm{dc}})}{(\delta_{\mathrm{dc}} - \delta_{\mathrm{nd+df,e}})(1 + \delta_{\mathrm{p}})} \text{ for } \delta_{\mathrm{nd+df,e}} < \delta_{\mathrm{p}} < \delta_{\mathrm{dc}} , \tag{5}$$

$$\beta_{\mathrm{dc}} = \beta_{\mathrm{p}} \text{ for } \delta_{\mathrm{p}} \geq \delta_{\mathrm{dc}} , \tag{6}$$

$$\beta_{\mathrm{dc}} = 0 \text{ for } \delta_{\mathrm{p}} \leq \delta_{\mathrm{nd+df,e}} , \tag{7}$$

$$\beta_{\mathrm{nd+df}} = \beta_{\mathrm{p}} - \beta_{\mathrm{dc}} . \tag{8}$$

The coarse dust depolarization ratio is given in Table 1.

Before we can proceed with step 2, we need to remove the coarse dust contributions to the particle depolarization ratio (Mamouri and Ansmann, 2014):

$$\delta_{\mathrm{nd+df}} = \delta_{\mathrm{p}} \text{ for } \delta_{\mathrm{p}} < \delta_{\mathrm{nd+df,e}} , \tag{9}$$

$$\delta_{\mathrm{nd+df}} = \delta_{\mathrm{nd+df,e}} \text{ for } \delta_{\mathrm{p}} \geq \delta_{\mathrm{nd+df,e}} . \tag{10}$$

In the following, $\delta_{\mathrm{nd+df}}$ is used with index e.

In step 2, we now can separate the fine dust backscatter coefficient $\beta_{\mathrm{df}}$ and the non-dust aerosol backscatter coefficient $\beta_{\mathrm{nd}}$,

$$\beta_{\mathrm{df}} = \beta_{\mathrm{nd+df}} \frac{(\delta_{\mathrm{dn+df}} - \delta_{\mathrm{nd}})(1 + \delta_{\mathrm{df}})}{(\delta_{\mathrm{df}} - \delta_{\mathrm{nd}})(1 + \delta_{\mathrm{dn+df}})} \text{ for } \delta_{\mathrm{nd+df}} > \delta_{\mathrm{nd}} , \tag{11}$$

$$\beta_{\mathrm{df}} = 0 \text{ for } \delta_{\mathrm{nd+df}} \leq \delta_{\mathrm{nd}} , \tag{12}$$

$$\beta_{\mathrm{nd}} = \beta_{\mathrm{nd+df}} - \beta_{\mathrm{df}} \tag{13}$$

with the backscatter coefficient $\beta_{\mathrm{nd+df}}$ (Eq. 8) for the mixture of non-dust and fine-dust particles, the respective linear depolarization ratio $\delta_{\mathrm{dn+df}}$ (Eqs. 9 and 10), the fine dust depolarization ratio $\delta_{\mathrm{df}} = 0.16$ for 532 nm, 0.21 for 355 nm, and 0.09 for 1064 nm according to Table 1, and the non-dust depolarization ratio $\delta_{\mathrm{nd}} = 0.05$.

### 4.3   Dust and non-dust extinction coefficients

Height profiles of dust extinction coefficients $\sigma_{\mathrm{d}}$, $\sigma_{\mathrm{df}}$, and $\sigma_{\mathrm{dc}}$ are obtained by multiplying the backscatter coefficients $\beta_{\mathrm{d}}$, $\beta_{\mathrm{df}}$,

and $\beta_{\mathrm{dc}}$ with respective lidar ratios, e.g., with $S_{\mathrm{d}} = 55$ sr for western Saharan dust at 355 and 532 nm (Tesche et al., 2009b,





2011a; Groß et al., 2011, 2015), of about 35-45 sr for eastern Saharan, Middle East and central Asian dust (Mamouri et al., 2013; Nisantzi et al., 2015; Hofer et al., 2017). If the non-dust aerosol component is of marine origin, as over Barbados during the summer season, we use a typical lidar ratio of $S_m = 20$ sr for marine particles at 355 and 532 nm and a slightly higher value of 25 sr for 1064 nm Gross2011, Rittmeister2017, Haarig2017b to obtain the marine particle extinction coefficient $\sigma_m$. If the non-dust aerosol is of continental (anthropogenic) origin, a reasonable lidar ratio would be $S_c = 50$ sr (at 532 nm) and 60-70 sr (at 355 nm) in the estimation of the extinction coefficient of anthropogenic aerosols $\sigma_c$.

For 1064 nm, directly observed particle lidar ratios for anthropogenic haze and biomass burning smoke are not available. We can only estimate them from combined lidar-photometer observations. The lidar delivers the vertically integrated backscatter coefficient (column backscatter) and the AERONET photometer yields the 1020 nm AOT. By using the wavelength dependence between the AOT values at 870 and 1020 nm, we may estimate the 1064 nm AOT by extrapolation. Finally, the ratio of the 1064 nm AOT and the 1064 nm column backscatter value provides an experimentally determined column lidar ratio. If we select combined lidar-photometer observations during dust dominated or pure marine aerosol conditions, we obtain reliable estimates of the respective 1064 nm dust and marine lidar ratio (Haarig et al., 2017b). In the same way, we can find anthropogenic haze lidar ratios during strong haze events. An example of the retrieval of the 1064 nm dust lidar ratio is given in Sect. 6.

## 4.4 Non-dust aerosol type

In case that the polarization lidar is equipped with a nitrogen Raman channel (at 387 and/or 607 nm) so that height profiles of the particle extinction coefficient at 355 and 532 nm can be determined (Rittmeister et al., 2017; Haarig et al., 2017a), we have the potential to evaluate whether the non-dust aerosol component is of marine or anthropogenic origin or even a mixture of both. The Raman lidar yields the particle extinction coefficient:

$$\sigma_p = \sigma_d + \sigma_m + \sigma_c. \tag{14}$$

Similarly, the POLIPHON method delivers estimates of the particle extinction coefficient by means of the following equation:

$$\sigma_p = S_d\beta_d + f_m S_m \beta_{nd} + f_c S_c \beta_{nd}. \tag{15}$$

$f_m$ and $f_c$ denote the relative contributions of marine and continental anthropogenic aerosol particles to the non-dust extinction coefficient $\sigma_{nd}$. In the case of a mixture of only two components (dust and smoke with $f_c = 1$ and $f_m = 0$ or dust and marine with $f_c = 0$ and $f_m = 1$), we can check by comparing the Raman-lidar extinction coefficient profiles (Eq. 14) and the POLIPHON extinction profiles (Eq. 15) whether the residual component is of marine or continental origin. In Sect 6, the best match of both extinction profiles is obtained for the Barbados lidar observations for $f_c = 0$, i.e., for the mixture of dust and marine particles. Ansmann et al. (2017) report an observation with a mixture of dust and marine particles in the lower part of the SAL and a mixture of dust and smoke in the upper part of the SAL.





### 4.5 Dust and non-dust particle mass concentrations

In the final step, the set of obtained particle backscatter and extinction coefficients are converted into particle volume concentration and by applying approriate values for particle density into mass concentrations. The mass concentrations $M_{df}$, $M_{dc}$, and $M_{nd}$ for fine dust, coarse dust, and non-dust particles (i.e., $M_m$, $M_c$), respectively, can be obtained by using the following

relationships (Ansmann et al., 2011a, 2012; Mamouri and Ansmann, 2014):

$$M_d = \rho_d c_{v,d,\lambda} \beta_{d,\lambda} S_{d,\lambda} \,, \tag{16}$$

$$M_{df} = \rho_d c_{v,df,\lambda} \beta_{df,\lambda} S_{df,\lambda} \,, \tag{17}$$

$$M_{dc} = \rho_{dc} c_{v,dc,\lambda} \beta_{dc,\lambda} S_{dc,\lambda} \,, \tag{18}$$

$$M_c = \rho_c c_{v,c,\lambda} \beta_{c,\lambda} S_{c,\lambda} \,, \tag{19}$$

$$M_m = \rho_m c_{v,m,\lambda} \beta_{m,\lambda} S_{m,\lambda} \,. \tag{20}$$

The particle densities $\rho_d$, $\rho_c$, $\rho_m$ for dust, continental aerosol pollution, or marine particles are assumed to be $2.6\,\mathrm{g/cm}^{-3}$, $1.55\,\mathrm{g/cm}^{-3}$, and $1.2\,\mathrm{g/cm}^{-3}$, respectively (Ansmann et al., 2012).

The extinction-to-volume conversion factors $c_{v,i,\lambda}$ applied to convert particle extinction coefficients into particle volume concentrations are listed in Tables 2 and 3 and studied in detail in Sect. 5. The overall uncertainties in all retrievals will be

discussed in Sect. 4.7. Standard deviations of all conversion parameters in Tables 2 and 3 are the basic information in the uncertainty analysis.

The dust-related conversion factors in Table 2 are derived from long-term as well as field campaign AERONET data sets. In the case of the Limassol (Cyprus) data, only observations with AE (440–870 nm) < 0.5 are considered. During SALTRACE-3 the 340 nm channel of the AERONET photometer was not working properly, so that we provide the respective values for

380 nm. For Cabo Verde and Barbados we computed the fine dust conversion factors $c_{v,df,\lambda}$ at 1064 nm from cases with fine-mode AOT>0.03, and for Cyprus we considered observations with fine-mode AOT>0.05 only in the calculations of $c_{v,df,\lambda}$ for 1064 nm. Otherwise, a strong impact of AERONET retrieval uncertainties become visible in our statistics. The values in Table 3 for anthropogenic haze and smoke are based on long term observations in Cyprus (Eastern Mediterranean) and Germany (Central Europe). Only observations with AE (440–870 nm) > 1.6 are used and interpreted as continental-aerosol-dominated

cases.

### 4.6 Consistency between one-step and two-step retrieval products

In Sect. 6, we will apply both, the one-step and the two-step methods to demonstrate that the entire POLIPHON concept is consistent, i.e., that the results obtained with both methods agree in terms of the profiles of the non-dust and total-dust backscatter coefficients. Such a consistency analysis is a necessary proof of the overall retrieval concept, but has not been

presented in our previous article (Mamouri and Ansmann, 2014).

By using the characteristic depolarization ratios in Table 1 and thus well defined values for $\delta_d$ and $\delta_{nd}$ in the one-step method and $\delta_{df}$, $\delta_{dc}$n and $\delta_{nd}$ in the two-step method, the only degree of freedom is the input profile of $\delta_{nd+df}$, i.e. the height profile of the




depolarization ratio for the mixture of non-dust and fine-dust particles. We can find the optimum set of one-step and two-step solution profiles by varying $\delta_{\mathrm{nd+df}}$ (for a given height level) until both solutions for, e.g., the non-dust backscatter coefficient (or the total-dust backscatter coefficient) are equal. We repeat this for each height level and obtain in this way the profile $\delta_{\mathrm{nd+df}}$. In Sect 6, we will present an example of this procedure. Such a consistency check can best be performed at comparable simple

and vertically homogeneous dust conditions, i.e., when the particle size distribution characteristics, and corresponding fine dust and coarse dust fractions are almost height-independent, which was usually the case over Barbados during the SALTRACE summer campaigns.

## 4.7 Retrieval uncertainties

Uncertainties in the basic lidar-derived optical properties and the POLIPHON retrieval products are extensively analyzed
and discussed by Freudenthaler et al. (2009), Tesche et al. (2009b, 2011a, b), Mamouri et al. (2013), Mamouri and Ansmann (2014, 2016), and Bravo-Aranda et al. (2016) and will not be discussed here. Typical uncertainties in the basic particle optical properties are given in Table 4 for 532 nm. The uncertainties for 1064 nm are simliar. However, in the case of the very uncertain 355 nm particle depolarization obtained with BERTHA (Haarig et al., 2017a) during SALTRACE, the errors may be at all a factor of two higher for the 355 nm POLIPHON products in Sect. 6. Uncertainties in the derived mass concentrations, also
given in Table 4, result from the uncertainties in the backscatter separation (as a function of the uncertainties in the input parameters), in the particle extinction coefficients, in the extinction-to-volume conversion factors (Tables 2 and 3), and in the assumed particle densities.

Haarig et al. (2017b) point to a another (new) uncertainty source in case that marine particles get mixed into layers with low relative humidity (RH) (below 50%). As long as RH is above 50% the particle depolarization ratio of the dried marine particles
is in the range of 0.05–0.08, and thus well enough covered by our assumption that non-dust aerosol causes a depolarization ratio of 0.05 or lower. However, if RH is 40% or even 20–30% in the SAL and marine particles get mixed into this layer they get completely dry and can cause depolarization ratios of the order of 0.15 at 532 nm and about 0.1 at 355 and 1064 nm. This may then be misinterpreted as contribution by dust particles. However, as long as the marine particle fraction (contribution to backscattering) in the SAL is low as during SALTRACE, this effect is negligible.

## 5 Extinction-to-volume conversion factors from AERONET

Trustworthy extinction-to-volume conversion factors $c_{\mathrm{v},i,\lambda}$ in Eqs. (16)–(20) are of key importance for an accurate retrieval of mass concentration profiles from lidar data (Ansmann et al., 2012; Mamouri and Ansmann, 2014). We determined these conversion factors from extensive correlation studies between the AOT and column-integrated particle volume concentrations obtained from long-term AERONET observations at Limassol, Cyprus, Leipzig, Germany, Ragged Point, Barbados, and the
mentioned short-term desert-dust field campaigns (AERONET, 2016). The results are shown in this section. Details of our AERONET data analysis can be found in Mamouri and Ansmann (2016).





AERONET-based correlation studies regarding the link between optical and microphysical aerosol properties were already presented in previous studies (Ansmann et al., 2011a, 2012). However these studies were performed by using few carefully selected AERONET observations (of the order of 5–20 cases) and covered only dust outbreak and urban haze situations. In previous papers we calculated coarse-mode-related conversion factors during dust outbreak situations and fine-mode-related

conversion factors during urban haze conditions, and applied them to lidar measurements of desert-dust/urban haze mixtures to separate the dust fraction and haze fraction.

Our recent analysis is based on 48 474 and 34 982 sun/sky photometer observation taken at Limassol and Leipzig, respectively. From these large data sets, we selected 421 Limassol and 974 Leipzig quality-assured data sets for the aerosol type of continental anthropogenic pollution (Ångström exponent AE>1.6). 125 observations with strong desert dust outbreaks and

optically dense dust layers are available from the AERONET observations during the SAMUM and SALTRACE field studies in Morocco, Cabo Verde, and Barbados. From the 2007–2015 Ragged Point AERONET observations we selected 123 observations of pure marine conditions. The selection criteria, which were applied to sort the individual observations into desert dust, anthropogenic pollution, and marine aerosol classes are defined in Mamouri and Ansmann (2016). In the following regression analysis we study the correlation between particle extinction and volume concentration for the defined aerosol types (marine,

dust, continental pollution). We performed the AERONET correlation study separately for all three laser wavelengths, but show the results for the mostly used lidar wavelength of 532 nm, only.

As shown in Mamouri and Ansmann (2016), the particle volume concentration $v$ is derived from the respective column volume concentrations and the extinction coefficient $\sigma$ is computed from the available AOT values. Interpolation between the measured AOTs at 8 wavelengths yielded the $\sigma$ values for the laser wavelengths. The correlations between $v$ and $\sigma$ at 532 nm

based on the desert dust field campaign observations is shown in Fig. 2. A clear linear dependence of volume concentration (fine, coarse, fine+coarse) on light extinction coefficient at 532 nm is given. The variability in the correlated data is very low. The respective correlation plots for pure marine conditions and for all three wavelength are presented in Fig. 3. The scatter in the plotted data is now much larger. Changes in the size distribution and thus in the correlation of $\sigma$ with $v$ can occur as a function of relative humidity in the MBL. Entrainment of free tropospheric air into the marine aerosol layer can sometimes lead

to relative humidities even below 50% in the upper half of the marine aerosol layer (Haarig et al., 2017b). At these conditions, marine particles dry and shrink and as a consequence change their optical properties and size characteristics in a less well defined way. Another reason may be that the uncertainties in the AERONET retrieval of the size distribution and volume concentration is likewise high when the AOT is very low, clearly below 0.1 for pure marine aerosol scenarios.

Figs. 4a and 4b shows the results of the correlations analysis for the long-term observations at Limassol and Leipzig for

urban haze scenarios.. Only cases with an Ångström exponent AE> 1.6 are considered. In addition, the correlation results for dust-dominated cases (AE<0.5) over Limassol are shown in Fig. 4c. The correlation between particle extinction coefficient and volume concentration for the fine-mode dominated continental aerosol observations is likewise high, keeping the potential impact of water uptake effects with changing relative humidity conditions, different sources for fine-mode particles, age, and transport ways (short for local and regional aerosol, long for remote sources) into account. The SD values of the continental

conversion factors are again of the order of 20% in Table 3





For all individual, single AERONET observations (belonging to a given marine, desert dust, or continental anthropogenic aerosol data set), we calculated $v/\sigma$ ratios for all three laser wavelengths. The mean values of $v_i/\sigma(\lambda)$ of aerosol type $i$ and given wavelength $\lambda$ are used as $c_{v,i,\lambda}$ in Eqs. (16)–(20). These mean values of $c_{v,i,\lambda}$ together with SD (obtained from the averaging procedure) are given in Tables 2 and 3. The SD values of coarse and total dust converison factors are around 10%.

For fine dust, marine and continental aerosol pollution, the SD values of the conversion factors are of the order of 20%.

## 6  Case study: SALTRACE observation on 20 June 2014

The POLIPHON method is applied to a triple-wavelength polarization observation of a well-defined SAL scenario with almost pure dust aerosol within the SAL (and a minor contribution of marine particles), and a vertically varying mixture of dust and marine aerosol below the African air mass in the marine aerosol layer over Barbados.

The selected SALTRACE case is described in detail in Sect. 3.2 in Haarig et al. (2017a) in terms of the measured particle optical properties. A short overview of the basic lidar observations of particle backscatter and depolarization-ratio profiles is given in Sect. 6.1. The solutions of the one-step POLIPHON data analysis separately performed for 355, 532, and 1064 nm are compared in Sect. 6.2. The results of the two-step method and the use of the one-step method to find the optimum set of solutions in terms of marine, fine dust, and coarse dust extinction and mass concentration profiles are presented and discussed

in the Sect. 6.3. In addition, the consistency between the Raman-lidar-derived total extinction coefficients with the sum of the extinction contributions by dust and non-dust (marine) aerosol components is shown. In Sect. 6.4, we finally apply the two-step POLIPHON method at all three wavelengths, check the consistency between all solutions, and discuss which of the wavelengths is most robust, i.e., most favorable to separate and retrieve fine dust, coarse dust, and non-dust aerosol extinction coefficient and mass concentration profiles.

### 6.1  Dust layering over Barbados on 20 June 2014

As shown in Figure 5, a 4 km deep Saharan dust layer was observed on 20 June 2014. The radiosonde profile of relative humidity shows a moist marine boundary layer (MBL) up to about 1.2 km height. Above the MBL, two dust layers were detected between 1.2 and 3 km height, and another layer from 3–4 km height. According to the backward trajectories in Fig. 6, African dust was transported on an almost direct way from western Africa to Barbados. The air mass needed 4–5 days to cross

the Atlantic Ocean before reaching the Caribbean. The thin dust layer between 3–4 km height crossed Saharan desert regions at heights around 6 km and thus contained only traces of dust. The lofted dusty air masses showed mostly relative humidities (RH) from 40–50%. Cirrus was present between 12 and 13.5 km height, and allowed us to check the quality of the triple-wavelength depolarization ratio observations (see Haarig et al., 2017a, for more details).

Figure 7 presents the mean profiles of the particle backscatter coefficient and linear depolarization ratio for all three laser

wavelengths for the three-hour period shown in Figure 5. The 532 nm particle depolarization ratio ranges from values of 0.25–0.27 below 2.4 km height and increases almost steadily from 0.25 at 2 km to 0.29 at the top of the SAL at 4 km height. The particle depolarization ratios at 355 and 1064 nm are lower with values from 0.22–0.26 than the ones at 532 nm and roughly





height-independent above 2 km height. The smaller 1064 nm depolarization ratios in the uppermost layer may indicate on average, smaller coarse-mode particles than in the main dust layer below 3 km height.

In the MBL, all depolarization ratios strongly decreases from about 0.25–0.27 at 1300 m height to values <0.1 at heights below 500 m above ground. These depolarization ratio values are still significantly above the marine depolarization ratio levels
of typically 0.02-0.03 and indicate significant downward mixing of dust over the island of Barbados.

## 6.2   1-step POLIPHON method

Figure 8 shows the dust extinction and dust mass concentration profiles obtained with the traditional dust/non-dust separation technique. The one-step retrieval method is applied separately to the lidar observations at 355, 532, and 1064 nm. The lidar ratio profiles for 355 and 532 nm obtained by applying the Raman-lidar method indicate typical western Saharan dust values from
50-60 sr (Tesche et al., 2009b, 2011a; Groß et al., 2011; Haarig et al., 2017a; Rittmeister et al., 2017). Below about 2.5 km height, the lidar ratios decrease to values around 40 sr (355 nm) and 45 sr (532 nm). This indicates the presence of marine particles in the lower dust layer which cause much lower lidar ratios (Groß et al., 2011; Rittmeister et al., 2017; Haarig et al., 2017b).

We used Eqs. (1)–(4) in Sect 4.1 to separate the dust and non-dust backscatter coefficients. For all three wavelengths, we
assumed that non-dust particles cause a depolarization ratio ≤0.05 and pure dust is indicated when the particle depolarization ratio exceeds 0.25 (355 nm), 0.31 (532 nm), and 0.27 (1064 nm). The obtained dust backscatter coefficients were then multiplied with lidar ratios of 55 sr (355 and 532 nm) and with 67 sr (1064 nm) to obtain the respective dust extinction coefficients shown in Fig. 8c.

The lidar ratio at 1064 nm was estimated from the AERONET observations of the 1020 nm AOT shortly before sunset
and the column backscatter coefficients at 1064 nm briefly measured after sun set. The AOT was 0.47-0.53 (1064 nm), 0.50–0.55 (532 nm), and 0.51–0.56 (355 nm). We subtracted the marine AOT (assuming a typical marine AOT of 0.06, 0.05 and 0.035 at 355, 532, and 1064 nm, respectively). AOT fine-mode fractions were 0.28, 0.2, and 0.06 for 355, 532, and 1064 nm, respectively. We considered slight changes in column backscatter values from before to after sun set as indicated by the continuous lidar observations and assumed that the same changes hold for the 1020 nm AOT. In this way we estimated the
1064 nm lidar ratio (ratio of 1020 nm AOT to 1064 nm column backscatter) to be close 67 sr, and, for consistency, we found a column lidar-ratio of 55 sr for the 500-532 nm wavelength range by using the 532 nm AOT from AERONET and the column backscatter coefficient for 532 nm from lidar.

Finally, the dust mass concentration profiles were calculated with Eq. (16) and the conversion factors in Table 2 (Cabo Verde, Barbados). The result is shown in Fig. 8d. The range of solutions (355–1064 nm) for the extinction coefficient and
mass concentrations in Figs 8c and d reflect well the uncertainties listed in Table 4. However, the uncertainty in the 355 nm particle depolarization ratios are very large and thus the 355 nm POLIPHON products in general have to be exercised with care. The strong positive deviations of the mass concentrations derived from the 1064 nm polarization lidar measurements compared to the respective profiles from the 532 nm polarization lidar observations are caused by the used conversion factor





of $0.73 \times 10^{-12}$ Mm. The actual 1064 nm conversion factor (for this day) was close to $0.65 \times 10^{-12}$ Mm as the AERONET observations showed.

### 6.3   2-step POLIPHON method

We start the discussion of the 2-step method with the most simple setup of input values for $\delta_{\text{df}}$, $\delta_{\text{dc}}$, and $\delta_{\text{nd+df,e}}$. We concentrate
on the lidar measurements at 532 nm. Besides the characteristic particle depolarization ratio for fine dust (0.16), coarse dust (0.39) and non-dust aerosol (0.05), a height-independent depolarization ratio $\delta_{\text{nd+df,e}}$ of 0.12 is assumed for the mixture of non-dust and fine dust. This corresponds to a backscatter contribution of 33% by non-dust particles (marine particles) and 67% by fine dust to the particle backscattering coefficient of the remaining aerosol (without the coarse dust backscatter contribution). Such an approach with height-independent $\delta_{\text{nd+df}}$ was already used in our previous arricle (Mamouri and Ansmann, 2014).

Figure 9 shows the result for the 20 June 2014 measurement. The dust and non-dust profiles obtained with the one-step method (already discussed in Fig. 8) and the two-step method are compared. As mentioned, in the first round of the two-step method (Eqs. 5–8), the coarse dust backscatter coefficient $\beta_{\text{dc}}$ is separated from the residual aerosol and in the second round (Eqs. 11–13), we separate the backscatter contributions of non-dust and fine dust. The results of the one-step and the two-step data analysis match well for the dust backscatter coefficients in the SAL (for heights >1.2 km height, best indicated by the
good agreement of the two non-dust backscatter profiles (blue profiles in Fig. 9b). However, within the marine boundary layer (<1.2 km height) with a changing mixing ratio of marine and dust particles, the solutions of the one-step and the two-step method are no longer in agreement. A fixed $\delta_{\text{nd+df,e}}$=0.12 is no longer appropriate at low heights. $\delta_{\text{nd+df}}$ decreases from 0.12 at 1.4 km height to 0.06 at 600 m height, where marine particle backscattering dominates.

In cases with changing mixtures of marine and dust aerosol particle, we need an alternative concept. The optimum approach
was described in Sect. 4.6 and is illustrated in Fig. 10. It makes use of the solutions of both, the one-step and the two-step methods, and searches for the most appropriate $\delta_{\text{nd+df}}$ in the two-step data analysis (Fig. 10b). The optimum $\delta_{\text{nd+df}}$ value is obtained, when the one-step dust (or alternatively the non-dust) backscatter values match with the respective two-step solutions (Fig. 10d). This computation is done separately for each height bin. At the end, a height profile of $\delta_{\text{nd+df}}$ is obtained (Fig. 10c) together with the profiles for non-dust, fine dust, and coarse dust backscattering in Fig. 10d. The non-dust backscatter fraction
in Fig. 10c shows that non-dust backscattering contributed to about 10–20% to total backscattering in the SAL. Below the SAL, the non-dust (marine) backscatter contribution rapidly increases towards 80% in the center of the marine boundary layer at 600 m height.

The mass concentration profiles in Fig. 10f shows that coarse dust mostly contributes to particle mass in the SAL and that the non-dust mass concentration contribution is negligibly small here. However, in terms of the radiatively and thus climate-
relevant particle extinction coefficient, the impact of fine dust is no longer insignificant in the SAL as shown in Fig. 10e.

Next, we checked the impact of a potential decrease of the coarse dust depolarization ratio from 0.39 to 0.35 as a result of gravitational settling of the larger dust particles which produce the highest depolarization ratios. This study can be regarded as a part of the uncertainty analysis. It should be emphasized again that the combined polarization lidar and AERONET photometer observations provide a unique and complementary set of information regarding the fine and coarse dust fractions





via the different fine and coarse dust depolarization characteristics (lidar) and the different fine and coarse dust contributions to AOT (sun photometer). Both concepts to separate fine and coarse mode fractions need to lead to a consistent set of fine-to-coarse dust backscatter and extinction ratios. For the SALTRACE case discussed here, the AERONET FMF values were close to 0.2 for this dust dominating case with an AOT close to 0.5 and a dust contribution of 90%. This means that the coarse mode

depolarization ratio was obviously close to 0.35 and not about 0.39. Only the combination of a fine dust depolarization ratio of 0.16 and and coarse dust depolarization ratio of 0.35 leads to the measured total dust depolarization ratio around 0.3 for a BFMF of 0.2.

Figure 11 now compares the results for both scenarios with BFMF=0.2 and 0.33 and the related coarse dust depolarization ratio of 0.35 and 0.39, respectively. As can be seen, a bias of 30-50% and around 20% must be taken into account in the profiles

of the retrieved fine and coarse dust profiles when using a coarse depolarization ratio of 0.35, and the true one is 0.39, or vice versa. These uncertainties are considered in Table 4. Note, that in Fig. 11b the one-step solution for the total mass concentration coincides (accidentally) with the coarse dust mass concentration profile obtained with the two-step POLIPHON method, so that only one profile is visible.

Finally, Figure 12 shows the attempt to identify the aerosol type causing the non-dust backscatter component (as explained

in Sect. 4.4). As can be seen, the profile of the total particle extinction coefficient obtained with the Raman lidar technique is in better agreement with the one obtained with the POLIPHON method for the mixture of dust and marine particles than for the mixture of dust and anthropogenic particles. In the uppermost dust layer from 3–4 km height, the non-dust contribution to extinction is rather low and it remains unclear whether the non dust component is of marine or anthropogenic origin.

The consistency between the solutions of the one-step and two-step data analysis as presented in Figs. 10 and 11 can best

be checked at these simplified aerosol conditions given over the remote tropical Atlantic. The retrieval procedure as illustrated in Fig. 10 is based on (and needs) the simplifying assumption that the ratio of fine to coarse dust backscattering is height-independent throughout the troposphere. This is a reasonable assumption for long-range-transported, aged dust layers, showing an almost height-independent particle depolarization ratio, as in Figs. 7, 9, and 10, thus suggesting invariant dust microphysical properties throughout the SAL. We further assume that this height-independent dust characteristics (height-independent fine-to-

coarse dust backscattering ratio) even hold for the layer below the SAL, into which the dust particles descend by gravitational setting or by turbulent downward mixing.

### 6.4 Combined 1-step 2-step POLIPHON method at triple wavelengths

In Fig. 13 the POLIPHON is now applied to all three lidar wavelengths. The goal is to check whether the full POLIPHON concept, tested in detail for 532 nm, is applicable to 355 and 1064 nm as well and to evaluate the overall consistency of the

entire set of solutions. The second question is: What is the best wavelength for the application of the POLIPHON method?

The results for 532 nm differ from the ones in Fig. 10 because of the assumption of a coarse dust depolarization ratio of 0.35 instead of 0.39. In this way the solutions for the optical properties are in agreement with the AERONET sun photometer observations which indicated a FMF of close to 0.2 (see discussion above). It should be mentioned again that a match of the solutions obtained with the one-step and two-step methods is only given for the backscatter coefficients so that the extinction




and mass concentration profiles can differ significantly because of the used lidar ratios and extinction-to-volume conversion factors.

The results can be summarized as follows. The overall agreement in Fig. 13 is good, especially for the main dust layer at heights above 2 km. Regarding the dust extinction coefficients, we obtained a reasonable, strong wavelength dependence for fine dust and a higher coarse dust extinction coefficient at 1064 nm than at 355 and 532 nm, which were found to be equal. Note again, that the 355 nm particle depolarization ratios were rather uncertain so that the 355 nm extinction coefficients and mass concentrations have much large error bars than the shown ones for 532 nm.

A very good agreement was found in the case of the mass concentrations for coarse and total dust, especially above 2 km height. Even in the case of fine dust (about an order of magnitude lower mass concentration compared to the coarse dust mass concentration values), a very good agreement is obtained for 532 and 1064 nm. Again, the high uncertainty in the 355 nm particle depolarization ratio may have prohibited a good match for all three wavelengths. In summary, Fig. 13 demonstrates that the retrieval could successfully be conducted for all three wavelengths.

The wavelength of 532 nm is probably the optimum POLIPHON retrieval wavelength because (a) the contrast between the fine and coarse dust depolarization ratio is highest (much higher than at 355 nm), which facilitates the separation of fine dust from coarse dust, (b) in contrast to 1064 nm, BFMF at 532 nm is high enough (the related BFMF is usually between 0.15 and 0.35) so that the fine dust backscattering has a significant impact on the measured lidar signals and can thus be separated with good accuracy. At 1064 nm, BFMF is around 0.06 so that an accurate retrieval of fine dust products is generally crucial. (c) The required profile of the 532 nm particle backscatter coefficient can be obtained with good accuracy which is a problem at 1064 nm because of the weak Rayleigh backscattering in the backscatter calibration height region. Finally, (d) at 532 nm, typically nitrogen Raman signals are measured in addition (as at 355 nm) so that aerosol typing of the non-dust component is possible. This option is not given at 1064 nm. For 532 nm, numerous lidar observations of the lidar ratio for all relevant aerosol types are available for the conversion of the dust and non-dust backscatter into the extinction coefficients. For 1064 nm, such observations are completely absent.

## 6.5 Discussion and summary

We demonstrated that the traditional, well-established one-step and the advanced two-step POLIPHON data analysis techniques are compatible and lead, in principle, to the same results in terms of mass concentrations, disregarding the used lidar wavelength (355, 532 or 1064 nm). However, the use of 532 nm signals has many advantages and leads to robust and most trustworthy products.

It is recommended to always use both (one-step and two-step) methods in the way shown above, although this approach may not be always fully applicable. It is restricted to conditions with height-independent dust size-distribution characteristics. These conditions may be best given in aged dust layers originating from desert dust outbreaks. At these simplified, dominating, and vertically homogeneous dust conditions we may even omit the use of the two-step method, and estimate fine and coarse dust fractions from the obtained total dust extinction coefficient (one-step approach). The optimum set of fine and coarse dust extinction profiles should then be in agreement with the AERONET observation of the fine-mode AOT fraction. The FMF



from the AERONET measurements holds for the total aerosol column. In less dust-dominated cases, one may use the lidar observations to determine the dust contribution to the total AOT and to estimate the dust-related FMF afterwards.

However, at complex aerosol conditions as describe by Mamouri and Ansmann (2014) and Nisantzi et al. (2014) with, e.g., lofted plumes of biomass burning smoke and injected soil dust, over a continental boundary layer with a mixture of, e.g.,
urban haze, dust, and marine particles, the two-step method is requested to identify the dust component and to estimate the fine and coarse dust contributions to the particle extinction coefficient and mass concentration. Examples of a combined use of the POLIPHON methodology together with sun photometer observations of AE and FMF was presented in Mamouri and Ansmann (2014). It was shown that the situation of soil-dust containing smoke plumes can be complicated by the fact that the coarse-mode may be widely depleted as found by Nisantzi et al. (2014) so that characteristic overall dust depolarization ratio of, e.g.,
0.31 at 532 nm, is no longer applicable. However, even in these complicated cases, comparisons of the results obtained with the one-step and two-step method are recommended to check the overall reliability of the found dust profiles.

One effect needs to be discussed, which can affect our dust separation approach in marine and coastal environments. As shown by Haarig et al. (2017b), marine particles get dry and become cubic in shape when the relative humidity changes from values of RH>80% to values <50%. At 40-50%, as in our case study at heights from 1.2–2.5 km, the depolarization may
increases from 2-3% in the humid MBL to 7–10% above the MBL. This increased particle depolarization will be misinterpreted as a contribution by dust, when assuming a non-dust depolarization ratio of 0.05, and thus will lead to an overestimation of dust extinction coefficient and mass concentration profiles. Our analysis showed that this dried marine depolarization effect is not very large in the case we discussed above, but needs to be kept in mind and needs to be further explored.

## 7   Conclusions

We used the unique opportunity of triple-wavelength polarization lidar observations during the SALTRACE field campaigns to test the recently introduced POLIPHON method in detail and extended the application to the three aerosol lidar wavelengths of 355, 532, and 1064 nm. The comparison of the products obtained with the traditional one-step method and the more advanced two-step technique showed the consistency of the overall POLIPHON concept. The extended POLIPHON technique is an important new approach to provide fine ($PM_{1.0}$) and coarse dust profiles ($PM_{10}$), and non-dust aerosol identification (marine or
aerosol pollution). Such detailed aerosol profiles are required to better estimate the impact of dust on climate and environmental conditions and to support respective atmospheric modeling and dust forecasting efforts (including assimilation of dust profile observations into the models). The technique is simple and robust and can be easily used in ground-based aerosol lidar networks and applied to spaceborne CALIOP lidar (532 nm wavelength) observations as well as in the case of the EarthCARE mission (with a high-spectral-resolution lidar operated at 355 nm).

As an open point, it remains to conduct comparisons of the lidar retrievals with respective in-situ observations which provide size distributions in the observed dust layers, and thus fine and coarse-mode-resolved volume and mass concentrations. More than 10 SALTRACE overflights with a research aircraft (Weinzierl et al., 2017) are available for this task. In April 2017, a dust





and aerosol-pollution-related field campaign with lidar and aircraft was conducted, this time in the highly polluted Eastern Mediterranean (Cyprus region), which offers another opportunity for intensive intercomparisons.

The new retrieval technique requires a considerable number of assumptions. The POLIPHON method critically relies on a good knowledge of depolarization ratios and lidar ratios for fine and coarse dust at 355, 532, and 1064 nm. Fortunately, meanwhile several laboratory and field studies exist which already helped to reduce the amount of made assumptions on the input parameters. Nevertheless, more laboratory studies (in future including even studies on fine and coarse-mode dust lidar ratios at the three main lidar wavelengths) are required to increase the overall accuracy of the POLIPHON products.

In addition, we need further simulation studies for the mentioned laser wavelengths and for realistic particle shape models. These efforts should focus on particle depolarization ratio and lidar ratios for fine dust and coarse dust, separately. Presently, we assumed that the lidar ratio for fine dust and for coarse dust is the same, models suggest, however, that the lidar ratio for fine dust is approximately a factor of two higher than the coarse dust lidar ratio (Gasteiger et al., 2011; Kemppinen et al., 2015a, b). The modeling activties have to be extended to characterize the light depolarizing features of coarse dust particles with diameters from 5–20 $\mu$m and related size parameters of 20 and higher.

## 8    Data availability

HYSPLIT backward trajectories are calculated via the available simulation tools (HYSPLIT, 2016). AERONET sun pho-tometer AOT data are downloaded from the AERONET web page (AERONET, 2016). SALTRACE BERTHA lidar data are available at TROPOS (e-mail: albert@tropos.de)

*Acknowledgements.* The perfect logistic support of CIMH during the SALTRACE preparation phase and intensive field phases in 2013 and 2014 is gratefully acknowledged. We are grateful to the SALTRACE lidar team of TROPOS at Barbados to provide us with unique polarization lidar observations. We thank the AERONET team here for high-quality sun/sky photometer measurements in Cyprus, Germany, Morocco, Cape Verde, and Barbados and respective the high quality data analysis. The authors acknowledge AERONET-Europe for providing calibration service. AERONET-Europe is part of ACTRIS-2 project that received funding from the European Union (H2020-INFRAIA-2014-2015) under Grant Agreement No. 654109. We would like to express our gratitude to the HYSPLIT team for the possibility to compute backward trajectories. We are grateful to Laurentiu Baschir and the Lidar Calibration Center (Lical), Bucharest, Romania, for excellent service regarding the characterization of lidar optical elements. This Lical activity is support by ACTRIS Research Infrastructure (EU H2020-R&I) under grant agreement no. 654169. We acknowledge funding from the EU FP7-ENV-2013 programme "impact of Biogenic vs. Anthropogenic emissions on Clouds and Climate: towards a Holistic UnderStanding" (BACCHUS), project number 603445.



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





**Table 1.** Fine and coarse dust linear depolarization ratios, $\delta_{df}$ and $\delta_{dc}$, respectively, as estimated from published laboratory studies (1:Järvinen et al. (2016), 2:Sakai et al. (2010)) and field observations (3:Freudenthaler et al. (2009), 4:Burton et al. (2015), 5:Veselovskii et al. (2016), 6:Haarig et al. (2017a), 7:Hofer et al. (2017)). 1064 nm coarse dust depolarization ratio can reach 0.4 close to dust sources (4).

| Wavelength | $\delta_{df}$ | $\delta_{dc}$ |
|---|---|---|
| 355 nm | 0.21±0.02 (1) | 0.27±0.03 (4,7) |
| 532 nm | 0.16±0.02 (1,2) | 0.37±0.03 (2,4,5,7) |
| 1064 nm | 0.09±0.02 (1) | 0.27±0.03 (3,4,6) |

**Table 2.** Extinction-to-volume conversion factors for desert dust required in Eqs. (16)–(18) in Sect. 4.5. The listed mean values and SD of $c_{v,i,\lambda}$ (in $10^{-12}$ Mm) are computed from averaging of all individual dust observations of these conversion factors of carefully filtered AERONET dust data sets (Cabo Verde and Barbados, Limassol, Cyprus). See the text in Sects. 4.5 and 5 for more details.

| Desert dust | $c_{v,d,\lambda}$ | $c_{v,df,\lambda}$ | $c_{v,dc,\lambda}$ |
|---|---|---|---|
| Cabo Verde, Barbados, 355–380 nm | $0.62 \pm 0.05$ | $0.15 \pm 0.02$ | $0.86 \pm 0.05$ |
| Cabo Verde, Barbados, 532 nm | $0.64 \pm 0.06$ | $0.21 \pm 0.04$ | $0.79 \pm 0.07$ |
| Cabo Verde, Barbados, 1064 nm | $0.73 \pm 0.06$ | $0.63 \pm 0.13$ | $0.72 \pm 0.04$ |
| Cyprus, 355 nm | $0.54 \pm 0.08$ | $0.15 \pm 0.04$ | $0.88 \pm 0.16$ |
| Cyprus, 532 nm | $0.61 \pm 0.07$ | $0.25 \pm 0.04$ | $0.81 \pm 0.15$ |
| Cyprus, 1064 nm | $0.76 \pm 0.09$ | $0.61 \pm 0.20$ | $0.72 \pm 0.08$ |





**Table 3.** Extinction-to-volume conversion factors for non-dust components (continental aerosol pollution, marine aerosol) required in Eqs. (19)–(20) in Sect. 4.5. The listed mean values and SD of $c_{v,i,\lambda}$ (in $10^{-12}$Mm) are computed from averaging of all individual marine or aerosol-pollution observations of these conversion factors of carefully filtered marine and aerosol-pollution AERONET data set (marine: Ragged Point, Barbados; aerosol pollution: Leipzig, Germany, Limassol, Cyprus). See the text in Sects. 4.5 and 5 for more details.

| Continental aerosol | $c_{v,c,\lambda}$ |
|---|---|
| Cyprus, 355 nm | $0.23 \pm 0.04$ |
| Cyprus, 532 nm | $0.41 \pm 0.07$ |
| Cyprus, 1064 nm | $1.41 \pm 0.32$ |
| Germany, 355 nm | $0.17 \pm 0.04$ |
| Germany, 532 nm | $0.30 \pm 0.08$ |
| Germany, 1064 nm | $0.96 \pm 0.34$ |
| Marine aerosol | $c_{v,m,\lambda}$ |
| Barbados, 355 nm | $0.53 \pm 0.12$ |
| Barbados, 532 nm | $0.65 \pm 0.14$ |
| Barbados, 1064 nm | $0.97 \pm 0.20$ |

**Table 4.** Typical uncertainties in the lidar-derived particle optical properties (for 532 nm wavelength), and volume and mass concentrations.

| Parameter | | Relative uncertainty |
|---|---|---|
| Backscatter coefficient | $\beta_p$ | 5–10% |
| Backscatter coefficient (desert dust) | $\beta_d$ | 10–15% |
| Backscatter coefficient (continental) | $\beta_c$ | 10–20% |
| Backscatter coefficient (marine) | $\beta_m$ | 20% (MBL) |
| Extinction coefficient (desert dust) | $\sigma_d, \sigma_{df}, \sigma_{dc}$ | 15–25%, 30–50%, 20–30% |
| Extinction coefficient (continental) | $\sigma_c$ | 20–30% |
| Extinction coefficient (marine) | $\sigma_m$ | 25% (MBL) |
| Mass concentration (desert dust) | $M_d, M_{df}, M_{dc}$ | 20–30%, 40–60%, 25–35% |
| Mass concentration (continental) | $M_c$ | 30–35% |
| Mass concentration (marine) | $M_m$ | 30–40% (MBL) |





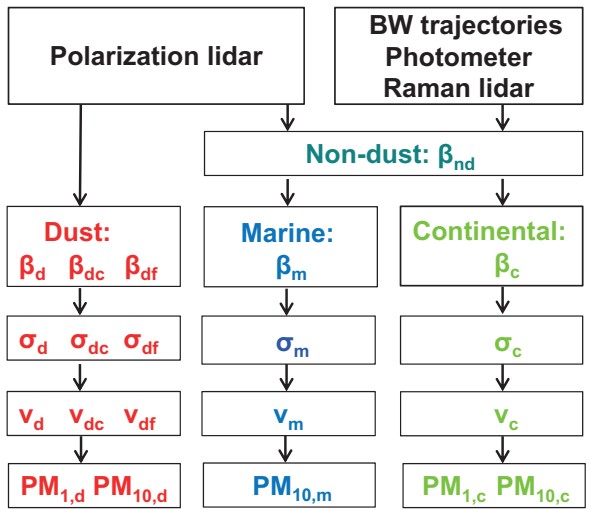

**Figure 1.** Overview of the POLIPHON data analysis. The depolarization ratio measurement enables us to separate the backscatter coefficients of fine dust, $\beta_{\mathrm{df}}$, coarse dust, $\beta_{\mathrm{dc}}$, and of the non-dust aerosol component, $\beta_{\mathrm{nd}}$. The non-dust backscatter coefficients are subsequently analyzed by using backward trajectories, auxiliary spectrally resolved photometer observations, and, if available, Raman-lidar observations of the total particle extinction coefficient to identify the non-dust aerosol type (marine or continental) and to quantify the contribution of marine particles ($\beta_{\mathrm{m}}$) and of the non-desert continental aerosol mixtures ($\beta_{\mathrm{c}}$) to the total particle backscatter coefficient. The backscatter coefficients $\beta_i$ are then converted to particle extinction coefficients $\sigma_i$, which in turn are converted to profiles of particle volume concentrations $v_i$. Finally, the respective particle mass (PM) concentrations for fine particles, $\mathrm{PM}_{1,i}$, and for fine + coarse particles, $\mathrm{PM}_{10,i}$, can be calculated. In the case of dust, we even determine $\mathrm{PM}_{10,\mathrm{d}}-\mathrm{PM}_{1,\mathrm{d}}$. Non-dust continental particles are mostly fine mode aerosol so that $\mathrm{PM}_{1,\mathrm{c}}$ is obtained, too.





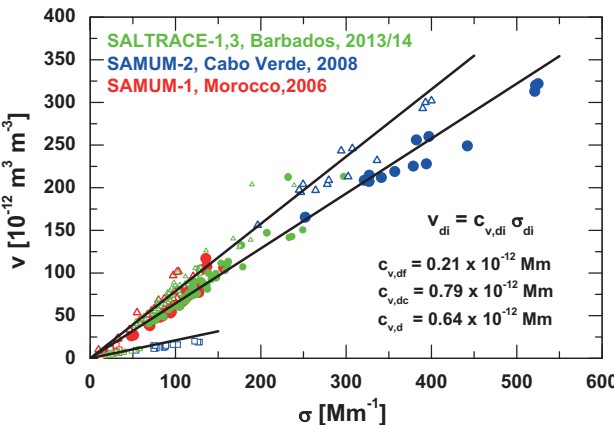

**Figure 2.** Relationship between dust layer mean 532 nm extinction coefficient $\sigma$ and particle volume concentrations $v$ considering only pronounced dust outbreaks observed during the SAMUM and SALTRACE campaigns in Morocco (red symbols, SAMUM-1, 2006), Cape Verde (blue, SAMUM-2, 2008), and Barbados (green, SALTRACE-1, 2013, SALTRACE-3, 2014). Correlations are separately shown for total (circles), fine-mode (squares), and coarse-mode (triangles) dust extinction ($\sigma_d$, $\sigma_{df}$, $\sigma_{dc}$) and respective volume concentration values ($v_d$, $v_{df}$, $v_{dc}$). The slope of the black lines indicate the mean increase of $v_d$, $v_{df}$, and $v_{dc}$ with $\sigma_d$, $\sigma_{df}$, and $\sigma_{dc}$. The respective extinction-to-volume conversion factors $c_{v,d}$, $c_{v,df}$, and $c_{v,dc}$ are given as numbers (also listed in Table. 2).

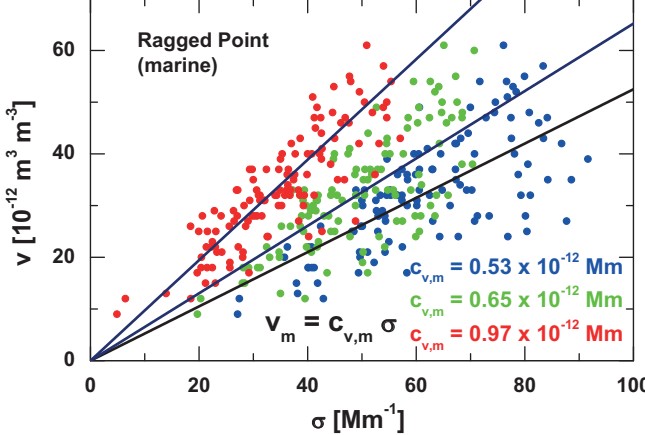

**Figure 3.** Relationship between particle extinction coefficient $\sigma_m$ and particle volume concentrations $v_m$ for pure marine cases measured at the AERONET station of Ragged Point, Barbados (2007-2015). Correlations are shown for 355 nm (blue), 532 nm (green), and 1064 nm extinction coefficients (red). The slope of the black lines indicate the mean increase of $v_m$ with $\sigma_m$. The respective conversion factors $c_{v,m}$ for the three laser wavelengths are given as numbers. They are also listed in Table 3.



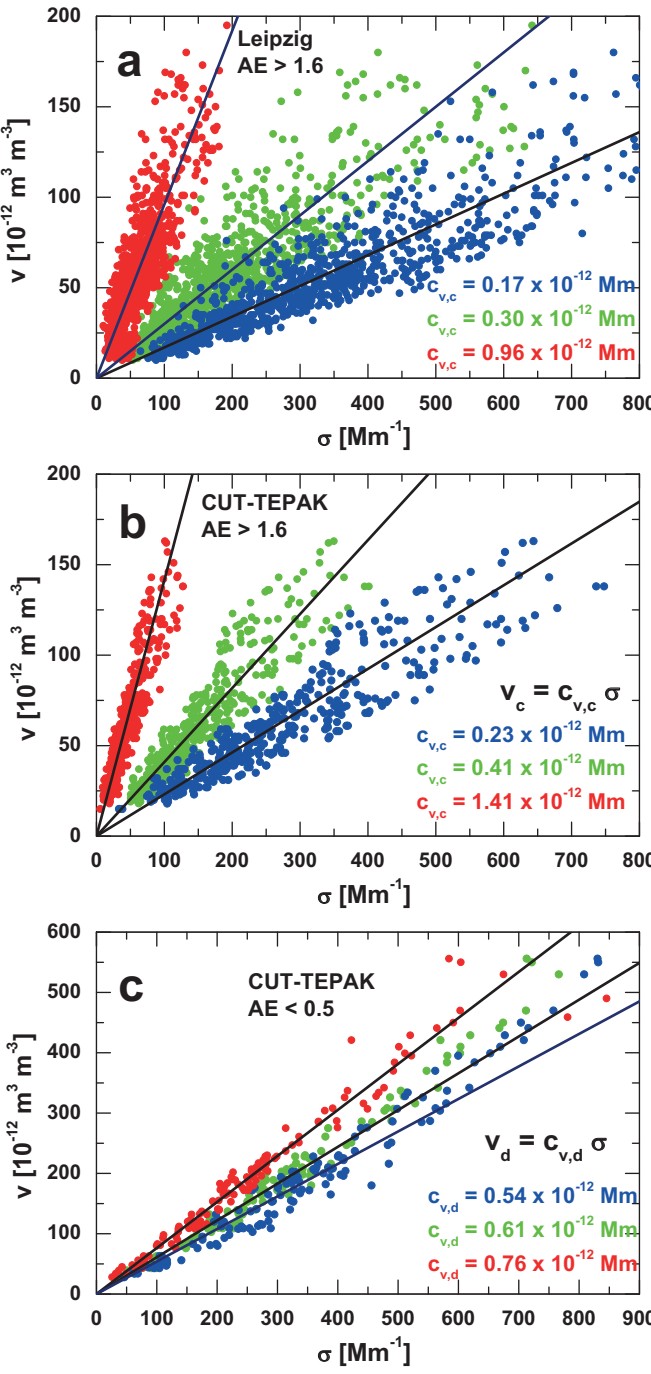

**Figure 4.** Same as Fig. 3, except for continental-aerosol-dominated cases measured at (a) Leipzig (only cases with Angström exponent, AE>1.6 are shown) and (b) Limassol (CUT-TEPAK AERONET site), and (c) for desert-dust-dominated cases at Limassol (AE<0.5). Correlations and mean conversion factors are shown for the laser wavelengths of 355 (blue), 532 (green), and 1064 nm (red). Conversion factors are given as numbers and also listed in Table 3.





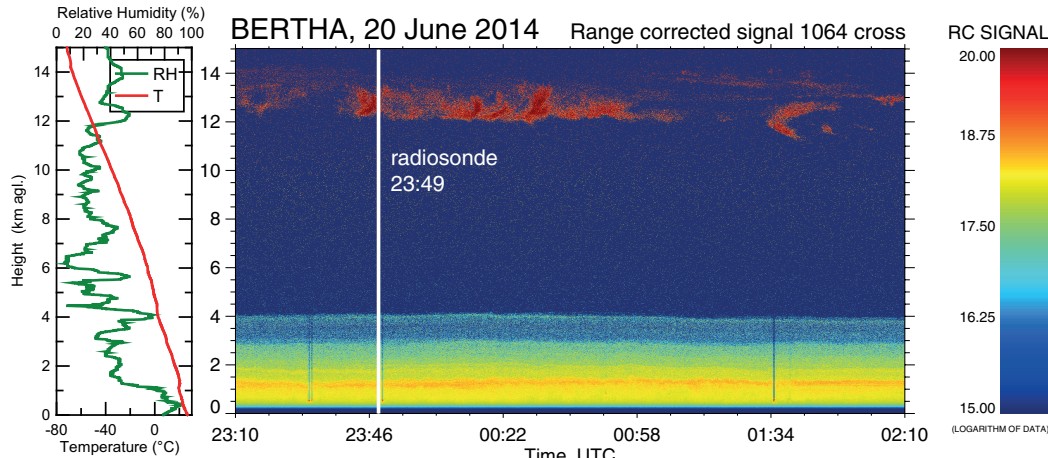

**Figure 5.** Desert dust layering over Barbados on 20 June 2014 (19:10–22:10 local time) during SALTRACE-3. The range-corrected cross-polarized 1064 nm lidar return signal is shown. Different dust layers are visible up to 4 km height. Highly depolarizing ice clouds were present between 12 and 14 km height. The cirrus layer was used to check the quality of the triple-wavelength depolarization ratio observations (Haarig et al., 2017a). The left panel shows radiosonde profiles of relative humidity (RH, green) and temperature (T, red). The radiosonde was launched at 23:49 UTC (indicated by a vertical white line in the right panel).

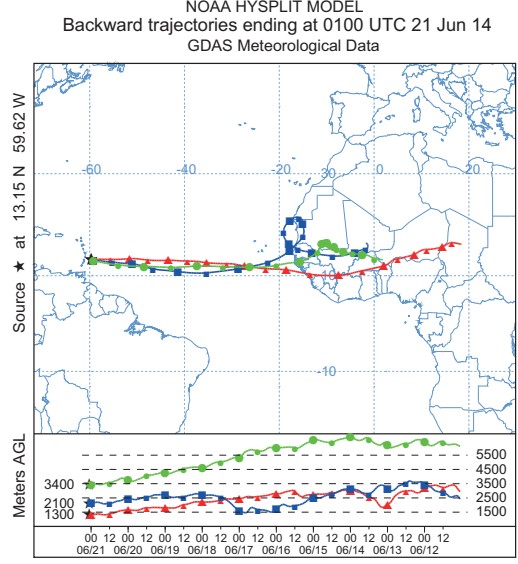

**Figure 6.** 10-day HYSPLIT backward trajectories for 21 June 2014, 01:00 UTC (HYSPLIT, 2016). The observed dust left the African continent four days before arrival over Barbados. Indicated arrival heights are 1.3 km (red), 2.1 km (blue), and 3.4 km (green).





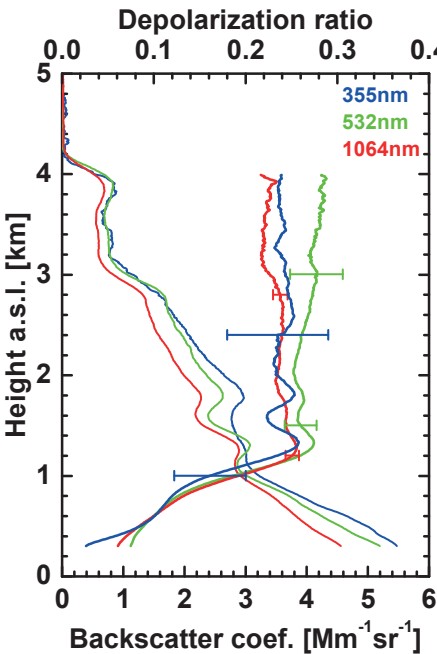

**Figure 7.** Triple-wavelength particle depolarization ratio profiling on 20 June 2014 during SALTRACE-3. Mean profiles of the particle backscatter coefficient (thin lines) and the particle linear depolarization ratio (thick lines) at 355 nm (blue), 532 nm (green), and 1064 nm (red) wavelength are shown. All signals in Fig. 5 are averaged (3-h period) and vertically smoothed with 200 m window length except in the case of the 355 nm depolarization ratio above 2.4 km height. Here, the signal smoothing length is 750 m (from 2.4–3.2 m height) and 1200 m (3.2–4.0 km height). Error bars indicate typical uncertainties in the depolarization ratio values (Haarig et al., 2017a). Relative errors of the particle backscatter coefficients are of the order of 5–10%.





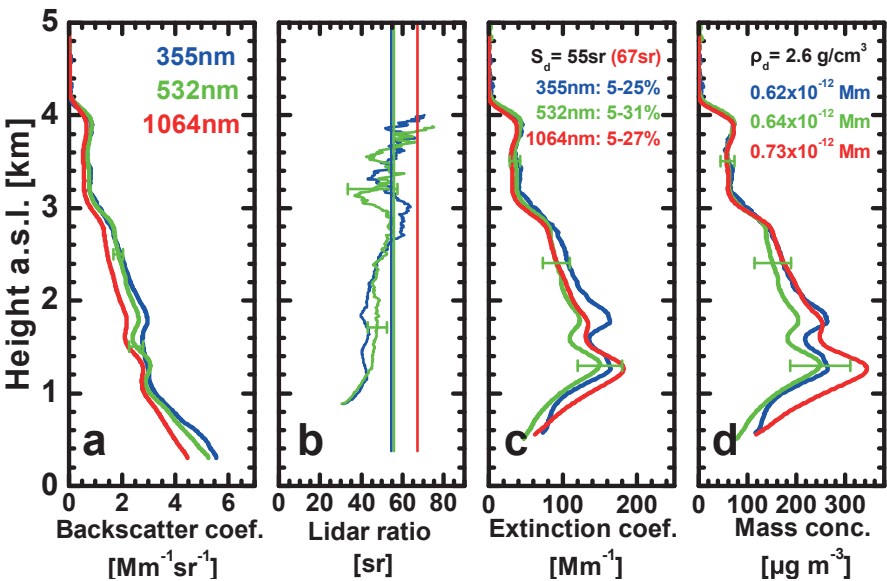

**Figure 8.** Dust extinction and mass concentration profiles (panels c and d) obtained by applying the one-step POLIPHON method to triple-wavelength polarization lidar observations. (a) Measured particle backscatter coefficients (as in Fig. 7), (b) Raman-lidar-derived extinction-to-backscatter ratios (for 355 nm in blue, for 532 nm in green) for the 3-h period shown in Fig. 5, (c) derived dust (only) extinction coefficients, and (d) dust mass concentrations. In the one-step POLIPHON data analysis (Sect. 4.1) we used characteristic non-dust (5%) and dust depolarization ratios (25, 31, and 27%, aslo given as numbers in panel c). The dust extinction coefficients are obtained by multiplying the dust backscatter coefficients with a lidar ratio of 55 sr (355, 532 nm) and 67 sr (1064 nm), see vertical lines in (b). The dust extinction coefficients are finally converted to dust mass concentrations by means of conversion factors in Table 2, given as numbers in panel d. We assume a dust particle mass density of 2.6 g/cm$^3$. Error bars show estimated overall retrieval uncertainties of 20% (extinction coefficient) and 25% (mass concentration) for 532 nm.



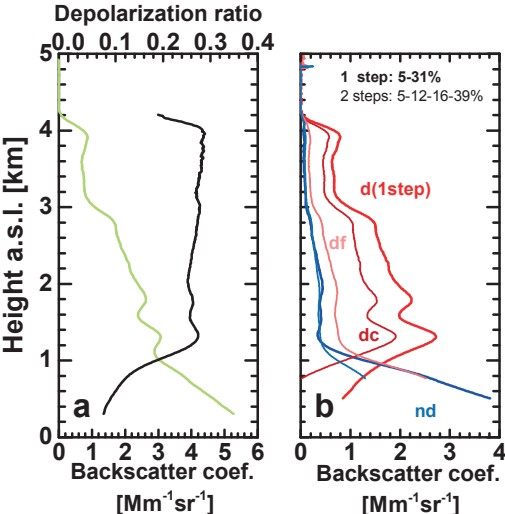

**Figure 9.** Dust backscatter profiles (red profiles in b) obtained by means of the one-step (d(1-step): total dust, Sect. 4.1) and the two-step POLIPHON method (df: dust fine mode, dc: dust coarse mode, Sect. 4.2) applied to 532 nm polarization lidar observations of the 532 nm particle backscatter coefficient (green in panel a) and particle linear depolarization ratio (black in panel a) as already shown in Fig. 7. The non-dust (marine) backscatter coefficient profiles (nd in panel b, one-step method, thick blue, two-step method, thin blue) are shown in addition. The one-step method makes use of the characteristic depolarization ratios of 5% (non dust) and 31% (dust), whereas the two-step method makes use of a height-independent non-dust plus fine-dust depolarization ratio $\delta_{dn+df}$ of 12% and coarse dust depolarization ratio $\delta_{dc}$ of 39% in the first round of the retrieval. In the second round, the non-dust depolarization ratio $\delta_{nd}$ of 5% and the fine dust depolarization ratio $\delta_{df}$ of 16% are used to separate marine and fine dust backscatter fractions .





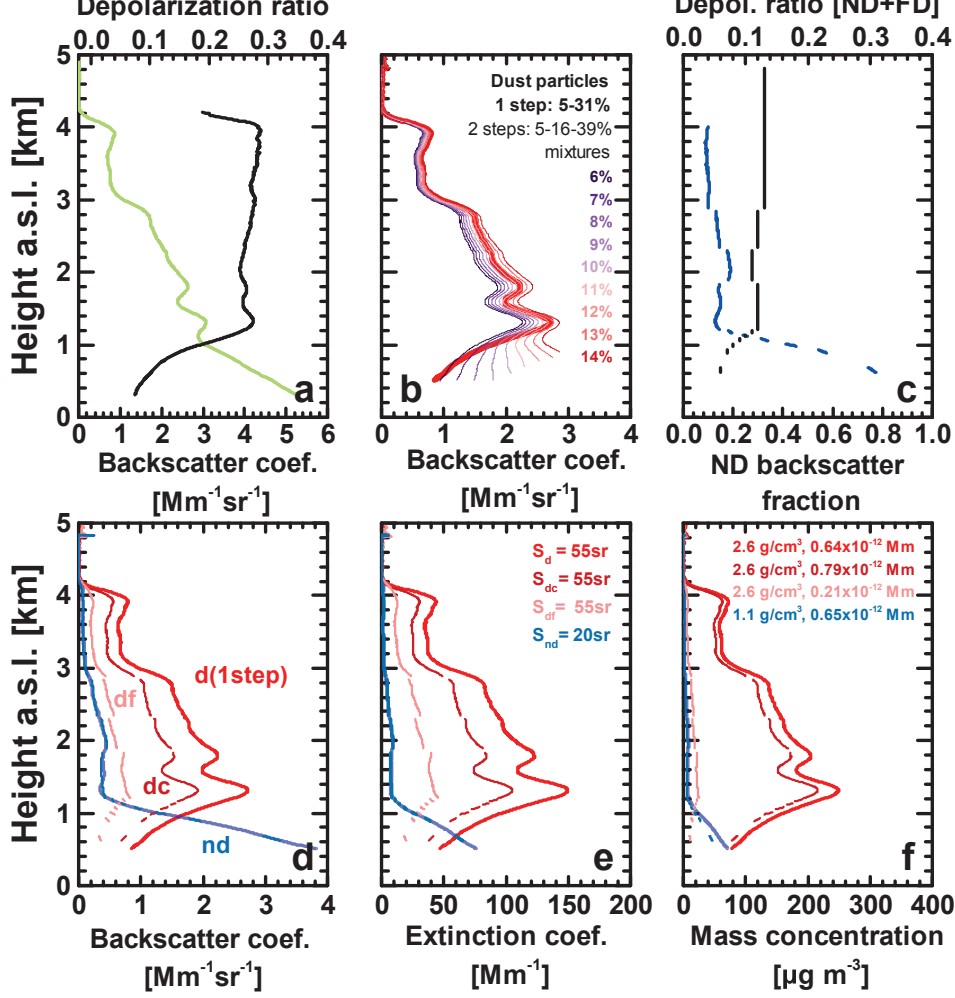

**Figure 10.** Combined use of the one-step and the two-step POLIPHON method to obtained optimum solutions for the dust backscatter and extinction coefficients (panels d, e) and dust mass concentration (panel f) separately for fine (df), coarse (dc), and total dust (d(1-step)). The respective non-dust (marine) profiles are shown in addition (solutions of one-step an two-step methods match perfectly). In (a), the measured 532 nm particle backscatter coefficient (green) and particle linear depolarization ratio (black, as in Figs. 7 and 8) are shown. In (b), the dust backscatter profile (one-step method, thick red) is compared with 10 profiles of the sum of fine and coarse dust backscatter coefficients (two-step method, thin lines) by assuming height-independent depolarization ratios $\delta_{dn+df}$ (for non-dust and fine dust) from 6 - 14% (in 1% steps) and a value for $\delta_{dc}$ of 39% for coarse dust in the first round, and depolarization ratios of $\delta_{nd}$ of 5% (non-dust) and $\delta_{df}$ of 16% (fine dust) in the second round of the two-step retrieval. In (c), the vertical distribution of the optimum non-dust plus fine-dust depolarization ratio $\delta_{dn+df}$ (black profile segments), for which the dust backscatter coefficients obtained with one-step and two-step methods match within ±0.05 Mm$^{-1}$ sr$^{-1}$ for each height bin. In addition, the corresponding profile of the non-dust-to-total particle backscatter ratio (ND backscatter fraction, blue profile segments) is shown. As in the foregoing figures, the dust extinction coefficients (in panel e) are obtained by multiplying the backscatter coefficients with the lidar ratio of 55 sr (dust, red) and 20 sr (marine particles, blue), and the total, fine, and coarse dust and marine particle mass concentrations (in panel f) are derived from the extinction profiles by using the conversion factors in Tables 2 and 3, and also listed as numbers in panel f together with the assumed particle densities.





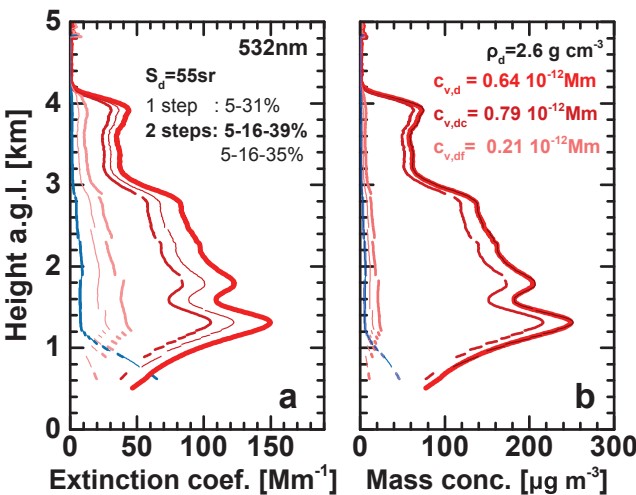

**Figure 11.** (a) Comparison of fine (light red) and coarse (dark red) dust 532 nm extinction profiles obtained with the 2-step method for two dust scenarios (scene 1: thick lines, BFMF=0.33, 2 steps: 5–16–39%, scene 2: BFMF=0.2, 2 steps: thin lines, 5–16–35%). (b) Respective mass concentrations, calculated from the extinction coefficients in (a). In both scenarios, the 1-step approach (very thick red line) uses the non-dust and dust depolarization ratios of 5% and 31%, respectively. The lidar ratio of 55 sr is applied to all (total, fine, coarse) backscatter coefficients to obtain the shown extinction profiles. The blue lines indicate the marine (non-dust) extinction and mass concentration profiles. A marine lidar ratio of 20 sr is used in the conversion of backscatter to extinction.





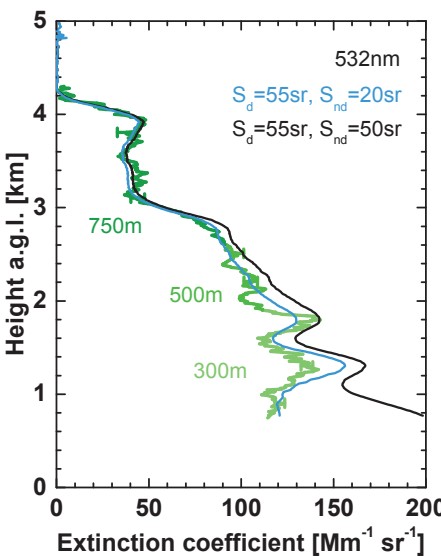

**Figure 12.** Comparsion of the particle extinction coefficient obtained with the Raman-lidar method at 532 nm and the POLIPHON extinction coefficient. In the case of the blue POLIPHON profile (for the mixture of dust and marine aerosol) the non-dust backscatter coefficient (in Fig. 10d) is multiplied by a marine lidar ratio of 20 sr and added to the dust backscatter contributions multiplied by a lidar ratio of 55 sr. The black curve (for the mixture of dust and anthropogenic aerosol) is obtained when the non-dust backscatter component in multiplied by a typical haze or smoke lidar ratio of 50 sr at 532 nm. To reduce the uncertainty by signal noise, signal smoothing with window lengths of 375 nm (up to 2.4 km height), 500 m (up to 3 km), and 750 m (above 3 km) was applied in the Raman lidar data analysis. Uncertainties are of the order of 10% (Raman lidar profile, green error bars) and 20% in the case of the POLIPHON solution.





**Figure 13.** Dust backscatter and extinction coefficients and mass concentrations for fine dust (FD), coarse dust (CD) and total dust (TD) obtained by means of the two-step POLIPHON method applied to all avaliable polarization lidar data sets (355 nm blue, 532 nm, green, 1064 nm, red). The results are in agreement with the respective products of the one-step method. For 532 nm the same solutions as in Fig. 10 are shown but now for BFMF of 0.2 and coarse dust depolarization ratio of 0.35. Error bars show the overall uncertainty in the case of the coarse and total dust profiles.