# Peer review of "Potential of polarization/Raman lidar to separate fine dust, coarse dust, maritime, and anthropogenic aerosol profiles"

_Atmospheric Measurement Techniques, 2017_

## Referee Comment (RC1) · Anonymous Referee #1 · 18 May 2017

The study presents an extended POLIPHON technique that allows the derivation of the mass concentration profiles from lidar measurements at three wavelengths and separating of the fine dust fraction, coarse dust fraction and non-dust contribution. The developed methodology is applied to SALTRACE observations and the results are presented. Also, the conclusions drawn from the analysis are clearly discussed: the use of the 532nm wavelength leads to more robust results, the analysis is better applied to conditions with height-independent ratio of fine to coarse dust backscattering. The paper is well conducted. The methodology part is clearly structured and explained. It is also well-stressed by the authors that the comparison of the results with in-situ

observations would greatly strengthen the retrievals of the proposed method.

Therefore, this is a great paper providing new paths for the retrieval of higher level lidar products (also applicable on space lidars) and I would therefore recommend publication with the following minor revisions:

For the estimation of extinction-to-volume conversion factors from AERONET, it is clearly stated that for the cases of continental and dust aerosol types (Germany and Cyprus sites), a filter has been applied based on the Angström exponent (i.e. AE>1.6 for continental and AE<0.5 for dust). Can you please be such specific also for the Barbados dataset concerning the marine type? Did you apply any filtering? Moreover, I wonder if the dataset obtained from AERONET in Cyprus contains also the relevant information regarding marine aerosol. In that case it would be beneficial for the manuscript to have a comparison with the corresponding values from Barbados.

Please provide some more details on Figure 10, I am confused. Especially in Figure 10b. You may want to consider adding more legends to adequately explain the method.

In Figure 2 you could use different colors for the different modes and different shapes for the different campaigns. Then you could plot with different colors also the corresponding correlation lines so that it would be more visible to the viewer

In Figures 3 and 4 I would recommend that you use lines with the corresponding colors for different wavelengths

In Figures 7, 9 and 11 the distinction between thick and thin lines is not at all visible to the reader. Try to make the difference between the lines greater or use dashed or dotted lines for the separation.

p.1, line 22: avoid 'and here' p.3, line 3: use 'except' instead of 'accept' p.3, line 24: add 'for the first time' p.5, line 32: "influenced" instead of "influences" p.9, line 3: add according 'to' p.16, line 9: 'article' instead of 'arricle' p.17, line 33: omit 'of' in the sentence 'a FMF of close to 0.2' p.18, line 7: reverse 'than the ones shown' p.19, line

3: write 'as described' p19, line 15: 'the depolarization increase' Fig.8 legend: 'also' instead of 'aslo' Fig. 12, legend: 'Comparison' instead of 'Comparsion'

---

## Referee Comment (RC2) · Anonymous Referee #2 · 1 Jun 2017

The manuscript by Mamouri and Ansmann describes an extension of the POLIPHON algorithm separating the contribution of fine-dust and coarse-dust aerosol particles and their mass concentration in the well-defined Saharan Air Layer as well as in the mixed boundary layer. The paper is well written, clearly structures and the method as well as the examples are described in a comprehensive and comprehensible way. Therefore I recommend publication after addressing only some minor comments.

General comments:

[Figure]

How is the derived conversion factor from AERONET affected by boundary layer aerosols?

The conversion factors of the continental aerosol over Cyprus and Germany show some differences. What may be the reason for these differences? Are you looking at different 'mixtures' at the different locations?

Specific comments:

P5, l11: How do you derive the BFMF of 0.06? Can you give some more details to that?

P5, l19ff: Can you clearly identify the article you refer to for the particle linear depolarization ratios at 532 and 1064 nm.

P8, l31 / P9, l1: Can you give a reference for the contribution of marine aerosols and fine dust? Are there any in-situ measurements available?

P10, l4: There is a typo concerning the references.

P 14, l21: As the boundary layer in this case is not only Saharan dust, I think you mean 4 km deep aerosol layer or 3 km deep dust layer?

---

## Author Comment (AC1) · 28 Jul 2017

Dear Reviewers!

We thank You for careful reading and providing many interesting and fruitful suggestions. They improved the paper.

**First of all, a short summary of the essential changes**

**1) Changes in the figures:**

**In Fig.2 (correlations of dust volume concentration vs dust extinction coefficient, separately for fine, coarse, total dust), we now include the Cyprus dust events. So, we have a top panel (a, SAMUM+SALTRACE data), and a (new!) bottom panel (b, Cyprus data).**

**Fig.3 now contains, in addition to the Barbados data, the pure marine scenarios observed with AERONET photometer at Cyprus (however only 9 cases in 4 years).**

**Figs 9, 10, and 11: The colors are now better to distinguish by selecting pink (fine dust), wine red (coarse dust), orange (total dust). Before we had just light red, red and dark red which was not easy to distinguish.**

**2) The entire text is re-checked, many parts are re-written to make the text more readable, complicated issues are rephrased as directly and indirectly requested by the reviewers.**

Step-by-step answers:

We only emphasis (in bold) the main and significant changes. Minor changes are not highlighted to facilitate reading.

Reviewer #1

For the estimation of extinction-to-volume conversion factors from AERONET, it is clearly stated that for the cases of continental and dust aerosol types (Germany and Cyprus sites), a filter has been applied based on the Angström exponent (i.e. AE>1.6 for continental and AE<0.5 for dust). Can you please be such specific also for the Barbados dataset concerning the marine type?

**We clearly state now in Sect. 5 (page 13) as well as in Figs. 2-4, how we define pure dust (500nm AOT > 0.1, AE<0.5) and pure marine scenarios (500nm AOT<0.07, AE<0.7).**

Did you apply any filtering?

**We simply checked all individual observations and selected just these cases which fulfill the requirement regarding AOT and AE, as mentioned above.**

Moreover, I wonder if the dataset obtained from AERONET in Cyprus contains also the relevant information regarding marine aerosol.

**Yes, we found 9 observations with 500nm AOT<0.07 and AE<0.7 withing the four-year data set. We found about 20 days with AOT < 0.08, but then AE was already 0.9 and higher. Cyprus is really a polluted region.**

In that case it would be beneficial for the manuscript to have a comparison with the corresponding values from Barbados.

**These 9 cases are now included in Fig.3 (pure marine aerosol scenarios, 123 Barbados observations, 9 Cyprus observations). The conversion factors (Barbados vs Cyprus) are rather similar and deviate by no more than 1% (355nm, 532nm), and 2% (1064nm). This is stated in Sect.5 (page 14).**

Please provide some more details on Figure 10, I am confused. Especially in Figure 10b. You may want to consider adding more legends to adequately explain the method.

**We reduced the number of curves in Fig.10b, from 10 to 4, and now these remaining curves (two step method) are in thin black, whereas the solution of the one-step method is in thick oranges. This contrast helps a lot to understand the (illustrated) retrieval. The retrieval details are now given in much more detail in Sect.6.3 (page 18).**

In Figure 2 you could use different colors for the different modes and different shapes for the different campaigns. Then you could plot with different colors also the corresponding correlation lines so that it would be more visible to the viewer.

**We follow the reviewer and changed all colors accordingly. Pink (fine dust), wine red (coarse dust), orange for total dust. All regression lines correspondingly in pink, wine red, and orange. Different AERONET sites are now indicated by different symbols. All in all, this much better now.**

In Figures 3 and 4 I would recommend that you use lines with the corresponding colors for different wavelengths.

**Done**

In Figures 7, 9 and 11 the distinction between thick and thin lines is not at all visible to the reader. Try to make the difference between the lines greater or use dashed or dotted lines for the separation.

**In Figure 7, now the depolarization ratio profiles are much thicker, in Figs. 9, 10, and 11, we use always the same colors, and we selected very contrasting colors, non-dust in blue, fine dust in pink, coarse dust in wine red, and total dust in orange.**

p.1, line 22: avoid 'and here'

**Sentence is rephrased.**

p.3, line 3: use 'except' instead of 'accept'

**Sentence is changed.**

p.3, line 24: add 'for the first time'

**time is added**

p.5, line 32: "influenced" instead of "influences"

**Done!**

p.9, line 3: add according 'to'

**Done!**

p.16, line 9: 'article' instead of 'arricle'

**Improved**

p.17, line 33: omit 'of' in the sentence 'a FMF of close to 0.2'

**Removed!**

p.18, line 7: reverse 'than the ones shown'

**Done!**

p.19, line 3: write 'as described'

**Done!**

p19, line 15: 'the depolarization increase'

**Done!**

Fig.8 legend: 'also' instead of 'aslo'

**Done!**

Fig. 12, legend: 'Comparison' instead of 'Comparsion'

**Done!**

Reviewer #2

General comments:

How is the derived conversion factor from AERONET affected by boundary layer aerosols?

**In Sect. 5 (page 14) we now discuss the interference of boundary layer aerosols (pollution, marine particles) on our correlation studies. We conclude that the impact of disturbing aerosols on the made dust correlations is always below 10%. Marine particles over Barbados do not really disturb because the size distributions of dust and marine aerosols are very similar and thus the correlations as well. Over Cyprus, the pollution of Limassol can affect the dust observations, but the restriction to AE <0.5 clearly selects the dust outbreaks, any significant impact of anthropgenic particles would cause AE of 0.8 and higher. So, the impact is again <10%. In the case of marine aerosols, we expect no interference, because we selected only observations with minimum AOT <0.07. In the case of pollution, the interference by dust or marine particles is expected to be low. The selection criteria of AE>1.6 should help to identify mostly cases with dominating aerosol pollution. Here, again we think that the interference is <10%, and thus of minor importance. All this is discussed briefly in Sect.5.**

The conversion factors of the continental aerosol over Cyprus and Germany show some differences. What may be the reason for these differences? Are you looking at different 'mixtures' at the different locations?

**This is now also explained in Sect. 5 (page 14). Limassol is always influenced by marine and dust particles, so there is always some coarse dust even if AE is high (>1.6). And Leipzig is far away from the Atlantic and the Baltic Sea and from deserts, but located in densely populated central Europe. So fine mode particles dominate, especially when AE is >1.6. This fits to the general dependence of the extinction-to-volume conversion factor which increases with mean size of the particle size by a factor of from fine mode aerosol to coarse dust. All this is now discussed in Sect.5 (page14).**

Specific comments:

P5, l11: How do you derive the BFMF of 0.06? Can you give some more details to that?

**We just report the FMF (AOT fine mode fraction) values as observed by AERONET. Fine and coarse mode AOTs are given in the data base for 440, 670, 870, and 1020nm. From these values we determine or estimated the dust FMF values in the UV (<400nm, 0.3-0.5), around 532 (0.15-0.3) and near infrared (0.05-0.08). We rephrased the respective paragraphs on page 5 to provide the message that we used AERONET data for FMF and then assumed FMF = BFMF (backscatter-related FMF).**

P5, l19ff: Can you clearly identify the article you refer to for the particle linear depolarization ratios at 532 and 1064 nm.

**We provide the reference now (Kemppinen et al., 2015b) in Sect.2.2 (page 5)**

P8, l31 / P9, l1: Can you give a reference for the contribution of marine aerosols and fine dust? Are there any in-situ measurements available?

**We rephrased the respective sentences to make clear that these are pure guesses (Sect. 4.2, page 9). We discuss… that we may use AERONET information on FMF and other information (climatological values for typical fine/coarse dust and marine particle ratios) to estimate the depolarization ratio for non-dust and fine dust. But we state that this estimation remains difficult. As a consequence, we searched for alternatives and introduce the comparison of dust backscatter coefficients obtained with the one step and the two-step retrievals, as described in Sect.4.6 and Sect 6 .3 (pages 17 and 18).**

P10, l4: There is a typo concerning the references.

**Improved!**

P 14, l21: As the boundary layer in this case is not only Saharan dust, I think you mean 4 km deep aerosol layer or 3 km deep dust layer?

**Improved!**

[revised manuscript text omitted]
_{\text{dc}} = \beta_{\text{p}} \frac{(\delta_{\text{p}} - \delta_{\text{nd+df,e}})(1 + \delta_{\text{dc}})}{(\delta_{\text{dc}} - \delta_{\text{nd+df,e}})(1 + \delta_{\text{p}})} \text{ for } \delta_{\text{nd+df,e}} < \delta_{\text{p}} < \delta_{\text{dc}} \,, \tag{5}$$

$$\beta_{\text{dc}} = \beta_{\text{p}} \text{ for } \delta_{\text{p}} \geq \delta_{\text{dc}} \,, \tag{6}$$

$$\beta_{\text{dc}} = 0 \text{ for } \delta_{\text{p}} \leq \delta_{\text{nd+df,e}} \,, \tag{7}$$

$$\beta_{\text{nd+df}} = \beta_{\text{p}} - \beta_{\text{dc}} \,. \tag{8}$$

The coarse dust depolarization ratio is given in Table 1. Eqs. (5)-(8) can be used to analyze the lidar observations separately for each of the three wavelengths.

Before we can proceed with step 2, we need to remove the coarse dust contributions to the particle depolarization ratio (Mamouri and Ansmann, 2014):

$$\delta_{\text{nd+df}} = \delta_{\text{p}} \text{ for } \delta_{\text{p}} < \delta_{\text{nd+df,e}} \,, \tag{9}$$

$$\delta_{\text{nd+df}} = \delta_{\text{nd+df,e}} \text{ for } \delta_{\text{p}} \geq \delta_{\text{nd+df,e}} \,. \tag{10}$$

In the following, $\delta_{\text{nd+df}}$ is used without index e.

In step 2, we now can separate the fine dust backscatter coefficient $\beta_{df}$ and the non-dust aerosol backscatter coefficient $\beta_{nd}$,

$$\beta_{df} = \beta_{nd+df} \frac{(\delta_{dn+df} - \delta_{nd})(1 + \delta_{df})}{(\delta_{df} - \delta_{nd})(1 + \delta_{dn+df})} \text{ for } \delta_{nd+df} > \delta_{nd}, \tag{11}$$

$$\beta_{df} = 0 \text{ for } \delta_{nd+df} \leq \delta_{nd}, \tag{12}$$

$$\beta_{nd} = \beta_{nd+df} - \beta_{df} \tag{13}$$

with the backscatter coefficient $\beta_{nd+df}$ (Eq. 8) for the mixture of non-dust and fine-dust particles, the respective linear depolarization ratio $\delta_{dn+df}$ (Eqs. 9 and 10), the fine dust depolarization ratio $\delta_{df} = 0.16$ for 532 nm, 0.21 for 355 nm, and 0.09 for 1064 nm according to Table 1, and the non-dust depolarization ratio $\delta_{nd} = 0.05$.

**4.3 Dust and non-dust extinction coefficients**

Height profiles of the dust extinction coefficients $\sigma_d$, $\sigma_{df}$, and $\sigma_{dc}$ are obtained by multiplying the backscatter coefficients $\beta_d$, $\beta_{df}$, and $\beta_{dc}$ with respective lidar ratios, e.g., with $S_d = 55$ sr for western Saharan dust at 355 and 532 nm (Tesche et al., 2009b, 2011a; Groß et al., 2011, 2015), of about 35-45 sr for eastern Saharan, Middle East and central Asian dust (Mamouri et al., 2013; Nisantzi et al., 2015; Hofer et al., 2017). If the non-dust aerosol component is of marine origin, as over Barbados during the summer season, we use a typical lidar ratio of $S_m = 20$ sr for marine particles at 355 and 532 nm (Groß et al., 2011; Rittmeister et al., 2017; Haarig et al., 2017b) and a slightly higher value of 25 sr for 1064 nm to obtain the marine particle extinction coefficient $\sigma_m$. If the non-dust aerosol is of continental (anthropogenic) origin, a reasonable lidar ratio would be $S_c = 50$ sr (at 532 nm) and 60-70 sr (at 355 nm) in the estimation of the extinction coefficient of anthropogenic aerosols $\sigma_c$. **In the case of lofted layers in the free troposphere with aged mixtures of dust and aerosol pollution (haze, smoke), the conversion of the backscatter into extinction profile may be complicated by the fact that the non-dust particles may grow during long-range transport (Müller et al., 2007) and respective changes in the non-dust lidar ratios cannot be excluded (Rittmeister et al., 2017).**

For 1064 nm, directly observed particle lidar ratios for anthropogenic haze, biomass burning smoke, mineral dust, and marine aerosols are not available. We can estimate them from combined lidar-photometer observations during aerosol conditions in which one of these aerosol components dominates the column optical properties. The lidar delivers the vertically integrated backscatter coefficient (column backscatter) and the AERONET photometer yields the 1020 nm AOT. By using the wavelength dependence between the AOT values at 870 and 1020 nm, we may estimate the 1064 nm AOT by extrapolation. Finally, the ratio of the 1064 nm AOT and the 1064 nm column backscatter value provides an experimentally determined column lidar ratio. An example of the retrieval of the 1064 nm dust lidar ratio is given in Sect. 6. Lidar ratios for 532 and 1064 nm for all main aerosol types can also be found in by Omar et al. (2009).

**4.4 Non-dust aerosol type**

In case that the polarization lidar is equipped with a nitrogen Raman channel (at 387 and/or 607 nm) so that height profiles of the particle extinction coefficient at 355 and 532 nm can be determined (Rittmeister et al., 2017; Haarig et al., 2017a), we

have the potential to evaluate whether the non-dust aerosol component is of marine or anthropogenic origin or even a mixture of both. The Raman lidar yields the particle extinction coefficient:

$$\sigma_{\mathrm{p}} = \sigma_{\mathrm{d}} + \sigma_{\mathrm{m}} + \sigma_{\mathrm{c}} \,. \tag{14}$$

Similarly, the POLIPHON method delivers estimates of the particle extinction coefficient by means of the following equation:

$$\sigma_{\mathrm{p}} = S_{\mathrm{d}}\beta_{\mathrm{d}} + f_{\mathrm{m}}S_{\mathrm{m}}\beta_{\mathrm{nd}} + f_{\mathrm{c}}S_{\mathrm{c}}\beta_{\mathrm{nd}} \,. \tag{15}$$

$f_{\mathrm{m}}$ and $f_{\mathrm{c}}$ denote the relative contributions of marine and continental anthropogenic aerosol particles to the non-dust extinction coefficient $\sigma_{\mathrm{nd}}$. In the case of a mixture of only two components (dust and smoke with $f_{\mathrm{c}} = 1$ and $f_{\mathrm{m}} = 0$ or dust and marine with $f_{\mathrm{c}} = 0$ and $f_{\mathrm{m}} = 1$), we can check by comparing the Raman-lidar extinction coefficient profiles (Eq. 14) and the POLIPHON extinction profiles (Eq. 15) whether the residual component is of marine or continental origin. In Sect 6, the best match of both extinction profiles is obtained for the Barbados lidar observations for $f_{\mathrm{c}} = 0$, i.e., for the mixture of dust and marine particles. Ansmann et al. (2017) report an observation with a mixture of dust and marine particles in the lower part of the SAL and a mixture of dust and smoke in the upper part of the SAL. The measurement was performed with a shipborne lidar (as part of SALTRACE) over the tropical Atlantic between Barbados and Cabo Verde during the final phase of the burning season in Africa in May 2013.

**4.5 Dust and non-dust particle mass concentrations**

In the final step, the set of obtained particle backscatter and extinction coefficients are converted into particle volume concentration and, by applying appropriate values for particle density, into mass concentrations. The mass concentrations $M_{\mathrm{df}}$, $M_{\mathrm{dc}}$, and $M_{\mathrm{nd}}$ for fine dust, coarse dust, and non-dust particles (i.e., $M_{\mathrm{m}}$, $M_{\mathrm{c}}$), respectively, can be obtained by using the following relationships (Ansmann et al., 2011a, 2012; Mamouri and Ansmann, 2014):

$$M_{\mathrm{d}} = \rho_{\mathrm{d}} c_{\mathrm{v,d},\lambda} \beta_{\mathrm{d},\lambda} S_{\mathrm{d},\lambda} \,, \tag{16}$$

$$M_{\mathrm{df}} = \rho_{\mathrm{d}} c_{\mathrm{v,df},\lambda} \beta_{\mathrm{df},\lambda} S_{\mathrm{df},\lambda} \,, \tag{17}$$

$$M_{\mathrm{dc}} = \rho_{\mathrm{dc}} c_{\mathrm{v,dc},\lambda} \beta_{\mathrm{dc},\lambda} S_{\mathrm{dc},\lambda} \,, \tag{18}$$

$$M_{\mathrm{c}} = \rho_{\mathrm{c}} c_{\mathrm{v,c},\lambda} \beta_{\mathrm{c},\lambda} S_{\mathrm{c},\lambda} \,, \tag{19}$$

$$M_{\mathrm{m}} = \rho_{\mathrm{m}} c_{\mathrm{v,m},\lambda} \beta_{\mathrm{m},\lambda} S_{\mathrm{m},\lambda} \,. \tag{20}$$

[revised manuscript text omitted]